

# Volcanic ash modeling with the on-line NMMB/BSC-ASH-v1.0 model: model description, case simulation and evaluation

**Alejandro Marti** [(1)(\*)], **Arnau Folch** [(1)], **Oriol Jorba** [(1)] **and Zavisa Janjic** [(2)]

[1]{Barcelona Supercomputing Center (BSC-CNS), Barcelona, Spain}

[2]{National Center for Environmental Prediction, College Park, Maryland, USA}

(*) Correspondence to: Alejandro Marti (Alejandro.Marti@bsc.es)

**Abstract**

Traditionally, tephra transport and dispersal models have evolved decoupled (off-line) from numerical weather prediction models. There is a concern that inconsistencies and shortcomings associated to this coupling strategy might lead to errors in the ash cloud forecast. Despite this concern, and the significant progress to improve the accuracy of tephra dispersal models in the aftermath of the 2010 Eyjafjallajökull and 2011 Cordón Caulle eruptions, to date, no operational on-line dispersal model is available to forecast volcanic ash. Here, we describe and evaluate NMMB/BSC-ASH, a new on-line multiscale meteorological and transport model that attempts to pioneer the forecast of volcanic aerosols at operational level. The model predicts volcanic ash cloud trajectories, concentration of ash at relevant flight levels, and the expected deposit thickness for both regional and global configurations. Its on-line coupling approach improves the current state-of-the-art of tephra dispersal models, especially in situations where meteorological conditions are changing rapidly in time, two-way feedbacks are significant, or distal ash cloud dispersal simulations are required. This work presents the model application for the first phases of the 2011 Cordón Caulle and 2001 Mt. Etna eruptions. The computational efficiency of NMMB/BSC-ASH and its application results compare favorably with other long-range tephra dispersal models, supporting its operational implementation.

**Keywords:** volcanic ash, on-line coupling, transport-meteorological modeling, operational forecast, NWPM, TTDM, Cordón Caulle, Mt. Etna.



## 1 Introduction

Explosive volcanic eruptions can eject large quantities of particulate matter (tephra) that, along with other aerosol droplets and trace gases, are carried upwards into the atmosphere by the buoyant eruption column and then dispersed by winds aloft (e.g. Sparks et al., 1997). Tephra particles smaller than 2 mm in diameter, technically defined as volcanic ash (Schmid, 1981), can spread over large distances away from the source forming ash clouds that jeopardize air-traffic (Casadevall, 1993), airports (Guffanti et al., 2009) and, for very large eruptions, alter both atmospheric composition and chemistry (Myhre et al., 2013; Self, 2006). Tephra Transport and Dispersal Models (TTDMs, e.g. Folch, 2012) are used to simulate the atmospheric transport, dispersion and ground deposition of tephra, and to generate operational short-term forecasts to support civil aviation and emergency management. The recent eruptions of Eyjafjallajökull (Iceland) in 2010 and Cordón Caulle (Chile) in 2011 have reinforced the importance of tephra dispersal in the context of global aviation safety. In addition to short-term forecast, other model applications include the reconstruction of past events, studying the impact of volcano eruptions on climate, probabilistic tephra hazard assessments or simulation of recent eruptions for model evaluation purposes. For any of those cases, TTDMs require a driving Numerical Weather Prediction Model (NWPM) or a meteorological reanalysis dataset for the description of the atmospheric conditions, and an emission or source model for the characterization of the eruption column (Fig. 1).

Traditionally, TTDMs have evolved decoupled (off-line) from NWPMs. In the off-line strategy, the meteorological driver runs *a priori* and independently of the TTDM to produce the required meteorological fields at regular time slabs (e.g. hourly). Meteorological data is then furnished to the TTDM, which commonly assumes constant values for the meteorological fields during each time slab or, at most, performs a linear interpolation in time. Although the off-line approach is operationally advantageous, there is a concern that it can lead to a number of accuracy issues (e.g. inaccurate handling of atmospheric processes) and limitations (e.g. neglect of feedback effects) that can be corrected by on-line approaches (Grell et al., 2004). These inconsistencies are especially important when meteorological conditions change rapidly in time or for long-range transport. However, uncertainties arising from off-line systems have received little attention, even if the experience from other communities (e.g. air quality) highlights the importance of coupling on-line dispersal and meteorological models (e.g. Grell and Baklanov, 2011). To date, only the Weather Research and Forecasting model with couple Chemistry (WRF-Chem; Grell et al., 2005) includes a coupled functionality that allows simulating emission, transport, dispersion, transformation and sedimentation of pollutants released during volcanic activities (Stuefer et al., 2012).

In this paper we describe and evaluate NMMB/BSC-ASH, a new on-line meteorological and atmospheric transport model to simulate the emission, transport and deposition of ash (tephra) particles released from volcanic eruptions. The model predicts ash cloud trajectories, concentration of ash at relevant flight levels, and the expected deposit thickness for both regional and global domains. The novel on-line coupling in NMMB/BSC-ASH allows solving both the meteorological and aerosol transport concurrently and interactively at every time-step. This coupling strategy aims at improving the current state-of-the-art of tephra dispersal models, especially in situations where meteorological conditions are changing rapidly in time, two-way feedbacks are significant, or distal ash cloud dispersal simulations are required. The model builds on the NMMB/BSC Chemical Transport Model (NMMB/BSC-CTM; Jorba et al., 2012; Pérez et al., 2011) to represent the transport



of volcanic particles. Its meteorological core, the Non-hydrostatic Multiscale Model on a B grid (NMMB; Janjic and Black, 2007; Janjic and Gall, 2012; Janjic, 2005; Janjic et al., 2011), allows for nested global-regional atmospheric simulations by using consistent physics and dynamics formulations. The final objective in developing NMMB/BSC-ASH is two-fold. On one hand, at a research level, we aim at studying the differences between the on-line/off-line modeling strategies. Moreover, a second version of the model is projected to quantify the feedback effects of dense volcanic ash clouds from large explosive eruptions on the radiative budget and local meteorology. On the other hand, at an operational level, the low computational cost of the NMMB dynamic core suggests that NMMB/BSC-ASH could be applied for more accurate on-line operational forecasting of volcanic ash clouds. Consequently, the focus on developing an on-line volcanic ash model is timely.

This manuscript is arranged as follows: Section 2 summarizes the modeling background and the standard physical schemes employed in NMMB/BSC-ASH; Section 3 provides a comprehensive description of the ash related modules, including details about the emission, transport, and deposition of volcanic particles; Section 4 validates the regional and global configurations of the model with simulations for the 2001 Mt. Etna and 2011 Cordón Caulle long-lasting eruptions; Section 5 discusses the implementation of the model for its operational use and; finally, Section 6 provides a summary conclusion of this work.

## 2    Modeling background

NMMB/BSC-ASH is a novel on-line multi-scale meteorological and atmospheric transport model developed at the Barcelona Supercomputing Center (BSC). The model attempts to pioneer the forecast of volcanic aerosols by embedding a series of new modules on the BSC's operational system for short/mid-term chemical weather forecasts (NMMB/BSC-CTM) developed at the BSC in collaboration with the U.S National Centers for Environmental Prediction (NCEP) and the NASA Goddard Institute for Space Studies. The development of the volcanic ash module follows the implementation of the mineral dust (Pérez et al., 2011) and sea-salt (Spada et al., 2013) modules in NMMB/BSC-CTM, and allows for a range of different physical parameterizations for research and operational use. The system allows for feedback processes among gases, aerosol particles and radiation, and includes a gas-phase module to simulate tropospheric gas-phase chemistry (Badia et al., 2016; Jorba et al., 2012).

Its meteorological core, the Non-hydrostatic Multiscale Model on a B grid (NMMB), is a fully compressible meteorological model with a non-hydrostatic option that allows for nested global-regional atmospheric simulations by using consistent physics and dynamics formulations. The standard physical and numerical schemes employed in NMMB are summarized in Table 1. The non-hydrostatic dynamics were designed to avoid over-specification.  The cost of the extra non-hydrostatic dynamics is about 20% of the cost of the hydrostatic part, both in terms of computer time and memory (Janjic, 2001, 2003). The numerical schemes for the hydrostatic and nonhydrostatic options available in the NMMB dynamic solver were designed following the principles found in Janjic (1977) and developed and modified thereafter (Janjic, 1979, 1984, 2003) and are summarized in Janjic and Gall (2012). The Arakawa B-grid horizontal staggering is applied in the horizontal coordinate employing a rotated latitude-longitude coordinate for regional domains and latitude-longitude





coordinate with polar filtering for global domains. In the vertical, the Lorenz staggering vertical grid is used with
a hybrid sigma-pressure coordinate. The general time integration philosophy in NMMB uses explicit schemes
when possible for accuracy, computational efficiency and coding transparency (e.g., horizontal advection), and
implicit for very fast processes that would otherwise require a restrictively short time-step for numerical stability
with explicit differencing (e.g., vertical advection and diffusion, vertically propagating sound waves). The
NMMB model became the North American Mesoscale (NAM) operational meteorological model in October of
2011, and it has been computationally robust, efficient and reliable in operational applications and pre-
operational tests since then. In high-resolution NWP applications, the efficiency of the model significantly
exceeds those of several established state-of-the-art non-hydrostatic models (e.g. Janjic and Gall, 2012).

## 3    The volcanic ash module: BSC-ASH

The BSC-ASH module is embedded within the NMMB meteorological model and solves the mass balance
equation for volcanic ash taking into account: i) the characterization of the source term (emissions); ii) the
transport of volcanic particles (advection/diffusion); and iii) the particle removal mechanisms
(sedimentation/deposition). The coupling strategy of BSC-ASH can be turned on or off, depending on the
solution required (on-line vs. off-line). While the on-line version of the model offers a more realistic
representation of the meteorological conditions, the off-line approach uses an "effective wind fields", which
aims to replicate the decoupling effect of off-line dispersal models, for which the wind velocity and mid-layer
pressure (sigma) are set to constant during a given coupling interval. The conservativeness of the model is
evaluated to ensure that the ash transport scheme is consistent with the mass conservation equation.

### 3.1    Source term

Explosive volcanic eruptions release large amounts of particles into the atmosphere. These particles, commonly
known as tephra, mix with ambient air to form an eruption column or volcanic plume. To forecast the ash cloud
movement and provide actual ashfall concentrations, tephra dispersal models require a complete characterization
of the parameters describing the source term. These parameters are generally referred to as Eruption Source
Parameters (ESPs) and include the eruption start and duration, column height, mass eruption rate (MER), vertical
distribution of mass and the particle grain size distribution (GSD). ESPs vary not only from one eruption to
another, but also during the different eruptive phases of a single event.
Typically, the eruption starting time, duration and column height are inferred/constrained from visual or satellite
observations. However, other parameters such GSD, MER, or the vertical distribution of mass in the column are
not available in real-time and must be inferred from previous events of similar characteristics (e.g. Mastin et al.,
2009). Uncertainties in source parameter values are a key factor limiting the accuracy of ash-cloud model
forecasts (Bonadonna et al., 2015a). The characterization of each ESP in NMMB/BSC-ASH is described in the
following subsections.





### 3.1.1 Mass eruption rate

The Mass Eruption Rate (MER) gives the mass released by unit of time and defines the eruption intensity. Its characterization in NMMB/BSC-ASH is achieved by employing a series of empirical correlations between (observed) column height and eruption rate, which, according to plume similarity theory, scales roughly as the 4[th] power of height. Because of this strong dependence, uncertainties within 20% in the determination of column height can translate into uncertainties up to 70% for the MER (e.g., Biass and Bonadonna, 2011). Averaged column heights of eruptions that have not been directly observed are typically derived from characteristics of tephra deposits (e.g. Bonadonna and Costa, 2013; Carey and Sparks, 1986; Pyle, 1989), or derived from model inversion (e.g. Connor and Connor, 2006; Pfeiffer et al., 2005).

The empirical correlations to estimate MER in the model are described in Table 2, and are based either on fitting observations (e.g. Mastin et al., 2009), or more sophisticated fits accounting for wind bent-over effects (e.g. Degruyter and Bonadonna, 2012; Woodhouse et al., 2013). In addition, MER can also be derived using a more sophisticated 1-D plume model (see Sect. 3.1.5).

### 3.1.2 Vertical distribution of mass

The vertical distribution of the initial column shape at the vent location is key when representing the plume, especially if wind shear exists with elevation at the volcano (Lin, 2012). To determine the vertical distribution of mass along the eruptive column, NMMB/BSC-ASH allows for the following geometrical distributions: i) point source, where mass is released as a single source point at height, $H_{plume}$; ii) top-hat, where mass is released along a umbrella-type slab of user-defined thickness, and iii) the so-called Suzuki distribution (Suzuki, 1983; Pfeiffer et al., 2005), which assumes a more complex vertical distribution of mass release along the eruption column;

$$S(z) = MER \left\{ \left(1 - \frac{z}{H_{plume}}\right) \exp\left[A\left(\frac{z}{H_{plume}} - 1\right)\right]\right\}^{\lambda}$$

(4)

where, $S$ is the mass per unit of time released at a given height $z$ above the vent, $MER$ is the total mass eruption rate, $H_{plume}$ is the column height, $A$ and $\lambda$ are the so-called Suzuki parameters. The parameter $A$ dictates the height of the maximum particle release (concentration), whereas $\lambda$ controls how closely mass distributes around this maximum. Any of the 3 options above can be combined independently with the different options for MER estimation. In NMMB/BSC-ASH, the terrain following hybrid sigma-pressure vertical levels of the model must be converted to elevations for each model integration time-step in order to interpolate $MER$ from the discrete source points into the nodes of the model grid.

### 3.1.3 Grain size distribution

The impact of explosive volcanic eruptions on climate and air traffic strongly depends on the concentration and grain size distribution (GSD) of pyroclastic fragments injected into the atmosphere (e.g. Girault et al., 2014).



Grain size distribution is normally reconstructed by volcanologists from grain size data at individual outcrops,
ranging from basic unweighted average of the GSD at individual sparse outcrops, to various integration methods
of grain size data (e.g. Rose and Durant, 2009). The particle grain size distribution in NMMB/BSC-ASH is
specified through an input file, which defines the particle bin properties (bin mass fraction, diameter, density and
shape factor). As typical in volcanology, grain size distributions are given in terms of the $\Phi$-number, defined as
$d = 2^{-\Phi}$, where $d$ is the particle diameter in mm. The granulometry file in the model can be furnished by the
user (typically derived from field data) or generated by an external utility program which produces Gaussian and
Bi-Gaussian distributions in $\Phi$ (log-normal in diameter $d$) (Costa et al., 2016; Folch et al., 2009).
### 3.1.4    Particle aggregation
The total grain size distribution (TGSD) erupted at the vent can be altered in case of particle aggregation, which
dramatically impacts particle transport dynamics thereby reducing the atmospheric residence time of aggregating
particles and promoting the premature fallout of fine ash. For computational purposes, particle aggregation in
NMMB/BSC-ASH is assumed to take place mainly in the eruption column, where particle concentration and
water contents are higher (the subsequent formation of aggregates downstream in the ash cloud under the
appropriate atmospheric conditions is not contemplated by the model). The model considers aggregates as
another particle class (bin), introduced as a standard source term by either solving: i) a series of simple analytical
expressions based on field observations or, ii) a more sophisticated wet aggregation model originally proposed
by Costa et al. (2010).
The analytical expressions available in the model modify the user-given particle grain size distribution by
assuming that a certain mass fraction of each granulometric class forms a new aggregate class added to the
TGSD. Despite the obvious limitations (obviates the physics of aggregation processes), these field-based
simplistic approaches are advantageous in that only the source term has to be modified in order to account for
aggregation. Table 3 provides an overview of these options. In addition to these empirical aggregation schemes,
NMMB/BSC-ASH also includes the wet aggregation model originally proposed by Costa et al. (2010). This
option allows for wet aggregation in the column providing an intermediate solution between the unaffordable all-
size class approach and the empirical solutions presented before. The model is based on a solution of the
classical Smoluchowski equation, obtained by introducing a similarity variable and a fractal relationship for the
number of primary particles in an aggregate. It also considers three different mechanisms for particle collision:
Brownian motion, ambient fluid shear, and differential sedimentation. Table 4 provides an overview of the
governing equations of this wet aggregation model.
### 3.1.5    FPlume model
A more sophisticated approach to obtain MER and the mass distribution in the column from the conditions at the
vent consists of solving a 1-D radially averaged BPT model for mass, momentum, and energy. These 1-D plume
models are more useful in operational roles and broad exploratory investigations (Costa et al., 2015; Devenish et
al., 2012). For that reason, NMMB/BSC-ASH is coupled with the 1-D FPlume model (Folch et al., 2015); a 1-D
cross-section averaged plume model which accounts for plume bent over, entrainment of ambient moisture,
effects of water phase changes on the energy budget, particle fallout and re-entrainment by turbulent eddies, as





well as variable entrainment coefficients fitted from experiments. The model also accounts for particle
aggregation in presence of liquid water or ice that depends on column dynamics, particle properties, and amount
of liquid water and ice existing in the column (Folch et al., 2010). This allows the plume model to predict an
"effective" grain size distribution depleted in fines with respect to that erupted at the vent. For a complete
definition of the governing equations of FPlume, refer to Folch et al. (2015). FPlume has two solving strategies
where the model: i) solves directly for column height for a given MER; or ii) solves iteratively for MER for a
given height. For any case, the following inputs need to be provided to the ash input file in NMMB/BSC-ASH:
eruption start and duration, vent coordinates and elevation, conditions at the vent (exit velocity, temperature,
magmatic water mass fraction, and total grain size distribution) and total column height or mass eruption rate.

## 3.2  Particle advection/diffusion

Transport of volcanic ash by advection and turbulent diffusion is analogous to those of atmospheric tracers (e.g.
moisture) transport (Janjic et al., 2009) in NMMB. Tracer advection is Eulerian, positive-definite and
monotonic. The Adams-Bashforth scheme is used for horizontal advection and the Crank-Nicholson scheme for
vertical advection. For the horizontal diffusion, the model uses a second order scheme with two types of
parameterized dissipative processes: explicit lateral diffusion (often called horizontal diffusion, a $2^{nd}$ order
nonlinear Smagorinsky-type approach; Janjic, 1990) and horizontal divergence damping (Janjic and Gall, 2012).
Plumes from high-intensity eruptions can be injected high into the stratosphere, reaching a maximum column
height and intruding laterally at the neutral buoyancy level (NBL) as a gravity current (Sparks et al., 1997). This
current can spread at velocities exceeding those of the surrounding winds, affecting tephra transport and
deposition near the source. As larger particles are removed by deposition and air is entrained, the plume density
decreases and momentum reduces such that, at a certain distance, atmospheric turbulence and wind advection
become the dominant atmospheric transport mechanisms (Baines and Sparks, 2005). Neglecting the gravitational
spreading of the umbrella cloud in tephra dispersal simulations could misrepresent the interaction of the volcanic
ash cloud and the atmospheric wind field for high-intensity eruptions and for proximal deposition of tephra
(Mastin et al., 2014). To account for the gravity-driven transport, NMMB/BSC-ASH is coupled with the model
of Costa et al. (2013) describing cloud spreading as a gravity current. This parameterization calculates an
effective radial velocity of the umbrella spreading as a function of time or cloud radius. The effective radial
velocity of the umbrella spreading is then combined with the wind field velocity centered above the vent in the
umbrella region to calculate the contribution of the gravitational spreading to the total cloud spreading. To
estimate the radial distance at which the critical transition between gravity-driven and passive transport occurs,
the umbrella front velocity is compared with the mean wind velocity at the NBL estimating the Richardson
number. Table 5 provides an overview of the governing equations of the gravity current model embedded in
NMMB/BSC-ASH.

## 3.3  Particle sedimentation and dry deposition

Particle sedimentation in NMMB/BSC-ASH is governed by the terminal velocity of sedimenting particles. This
fall velocity is sensitive to particle size and atmospheric conditions, determining the residence time of ash


particles in the atmosphere. The NMMB/BSC-CTM model assumes that the settling velocities of aerosols
(mineral dust, sea salt, etc.) follow the Stokes law for spherical particles corrected by the Cunningham slip
factor. The Stokes law applies to the creeping or Stokes flow regime, in which the drag force is proportional to
particle velocity, and holds only for Reynolds numbers Re≤0.1. This method is considered an efficient removal
mechanism for small particles (< 20 µm). Therefore, calculated fallout times based on settling according to
Stokes Law are inaccurate for coarse ash (> 64 µm), which sediments much faster. In addition, ash particles are
not spherical, which complicates and further slows fallout. In order to simulate a wider spectrum of particle
sizes, NMMB/BSC-ASH adds a new sedimentation module that covers the turbulent regime (Re≥1000) in where
the drag force is proportional to the square of the particle velocity. In this case, the particle settling velocity, $v_s$,
can be expressed as:

$$v_s = \sqrt{\frac{4g\left(\rho_p - \rho_a\right)d}{3C_d\rho_a}}$$

(13)

where, $\rho_a$ and $\rho_p$ denote air and particle density, respectively, $d$ is the particle equivalent diameter, and $C_d$ is the
drag coefficient (depending on the Reynolds number). Strictly, the expression above is valid for spherical
particles in the turbulent regime but it is often generalized to the whole range of Re numbers and particle shapes
by defining the drag coefficient properly. Table 6 provides an overview of the different settling velocity models
available in NMMB/BSC-ASH, each relaying on different empirical evaluations of drag coefficient.
Dry deposition, acting at the bottom layer of the model, is a complex process depending on physical and
chemical properties of the particle, the underlying surface characteristics and micro-meteorological conditions.
Dry deposition in NMMB/BSC-ASH is based on that originally proposed by Zhang et al. (2001). This
parameterization has been updated to account for the different settling velocities available for volcanic particles -
Eq. (13). The dry deposition velocity in the model, $v_d$, is given by:

$$v_d = v_s + \frac{1}{(R_a - R_s)}$$

(18)

where, $R_a$ is the aerodynamic resistance of the particle, and $R_s$ is the surface resistance. It is worth mentioning
that, for most of its resident time, airborne volcanic ash lies above the near-surface atmospheric layers, where
gravitation dominates, implying that, in most cases, dry deposition has little influence on model results.
**3.4    Mass conservation**
Mass conservation is a critical requirement for any atmospheric transport algorithm. Non-conservative schemes
can significantly underestimate or overestimate concentrations, especially for long time integrations, in which it





is critical that the tracer advection scheme is consistent with the mass continuity equation (Jöckel et al., 2001).
Most mesoscale meteorological models use observation/analyzed fields or global model results as initial
conditions, and therefore they are not very sensitive to slowly accumulated mass inconsistencies as re-
initializations remove accumulations. However, dispersal models are usually very sensitive to mass
inconsistencies set in previous simulations or spin-up fields as initial conditions, thereby accumulating mass
inconsistencies. In addition to mass conservation, monotonicity and prevention of non-physical under and
overshoots in the solution are also a highly desirable characteristics in transport schemes (Rood, 1987). For these
reasons, the model includes a conservative, positive definite and monotone Eulerian scheme for advection. The
positive definiteness in the model is guaranteed by advecting the square root of the tracer using a modified
Adams-Bashforth scheme for the horizontal direction and a Crank-Nicholson scheme for the vertical direction.
The conservation of the tracer is achieved as a result of the conservation of quadratic quantities by the advection
scheme. Monotonization is applied *a posteriori* to eliminate new extrema (Janjic et al., 2009). The conservative
nature of NMMB/BSC-ASH is evaluated by calculating the mass flux at the boundaries (for regional domains)
of the computational domain, the airborne mass, and the mass deposited on the ground to verify mass
conservation at each time-step (e.g. < 0.5% mass creation for a 30 day simulation).
**3.5    Numerical performance**
The high computational efficiency of the NMMB meteorological driver allows for the application of
nonhydrostatic dynamics at a global scale (Janjic et al., 2009), and supports that the NMMB/BSC-ASH could be
used in an operational forecast of volcanic ash clouds. Model parallelization is based on the well-established
Message Passing Interface (MPI) library. The computational domain is decomposed into sub-domains of nearly
equal size in order to balance the computational load, where each processor is in charge to solve the model
equations in one sub-domain. The Eulerian schemes in the model require relatively narrow and constant width
halos, which simplify and reduce communications.
To measure the time-to-solution required, we compute the parallel speed-up (computation speed) of the model;
that is, the performance gains of parallel processing in comparison to serial processing:

$$S_{(P)} = \frac{t_{(P=1)}}{t_{(P)}} \tag{19}$$

where $S$ is the computed speed-up value, and $t$ is the simulation run-time employing $P$ processors instead of
running it serially ($P = 1$).
To evaluate the efficiency of the model while using the computational resources, the parallel efficiency of the
model is computed by looking at the ratio between the parallel speed-up over $P$:

$$E_{(P)} = \frac{S_{(P)}}{P} \tag{20}$$



Parallel efficiency is used as a metric to determine how far the model's speed-up is from the ideal. If the speed-
up is ideal, the efficiency is 1, regardless of how many cores the program is running on. If the speed-up is less
than ideal, the efficiency is less than 1.
**4      Simulations and validation**
The forecast skills of NMMB/BSC-ASH have been tested for several well-characterized eruptions, including the
Pinatubo 1991 (Philippines), Etna 2001 (Italy), Chaitén 2008 (Chile) or Cordón Caulle 2011 (Chile) eruptions.
Here, we present two applications of the model for the ash dispersal forecast of weak and strong long lasting
eruptions. Section 4.1 summarizes the results of the regional and global simulations for the first days of the 2011
Cordón Caulle eruption. This event represents a suitable case study of strong long-lasting eruptions, which is
useful when evaluating the on-line coupling strategy of the model. In a parallel effort, Sect. 4.2 summarizes the
results from the regional configuration of the model for the 2001 Etna eruption. This eruption is a good example
of weak long-lasting eruptions, useful when evaluating the sedimentation mechanisms of the model against well-
characterized tephra deposits.
**4.1      The 2011 Cordón Caulle eruption**
The 2011 Cordón Caulle eruption was a typical mid-latitude Central and South Andean eruption, where
dominating winds carried ash clouds over the Andes causing abundant ash fallout across the Argentine
Patagonia. Besides the significant regional impacts on agriculture, livestock and water distribution systems, this
eruption stranded thousands of passengers due to air traffic disruptions in the southern hemisphere, thereby
causing important economic losses to airlines and society (e.g. Raga et al., 2013; Wilson et al., 2013). This event
is evidence of the global nature of the volcanic ash dispersion phenomena and highlights the need for accurate
real-time forecasts of ash clouds.
The Cordón Caulle volcanic complex (Chile, 40.5º S, 72.2º W, vent height 1420 m a.s.l.) reawakened on 4 June
2011 around 18:30 UTC after decades of quiescence. The initial explosive phase spanned over more than two
weeks, generating ash clouds that dispersed over the Andes. The climatic phase (~27 h) (Jay et al., 2014) was
associated with a ~9 km (a.s.l.) high column (Osores et al., 2014). For the period between 4 - 14 June, numerous
flights were disrupted in Paraguay, Uruguay, Chile, southern Argentina and Brazil. The two major airports
serving Buenos Aires and the international airport in Montevideo, Uruguay, were closed for several days, along
with airports in Patagonia (Wilson et al., 2013). A detailed chronology of the eruption can be found in Collini et
al. (2013), the stratigraphy and characteristics of the resulting fallout deposit are described in Pistolesi et al.
(2015) and Bonadonna et al., (2015b), and a summary of the environmental impacts of the eruption is discussed
in Raga et al. (2013) and Wilson et al. (2013).
Figure 2 shows the synoptic meteorological situation during the 6-16 June. Here, we give a brief chronology of
the events that occurred during the first two weeks of eruptive activity in order to compare them with the
predictions of the model. In particular, we focus on the three major dispersion episodes occurring between the 6 -




8 June. The first major episode, on 6 June, resulted in a cloud moving northwards at high atmospheric pressure
(300 hPa), reaching the northern regions of the Argentine Patagonia threating the Buenos Aires air space. This
resulted in major air traffic disruption at the two international airports that service the city: Aeroparque (AEP)
and Ezeiza (EZE), which remained closed intermittently during the following 15 days. A complementary episode
dispersed ash further to the north of Argentina leading to a more recognizable shift of winds over the E-SE. In
the morning of 7 June, the initial trough reached the northern boundary of Paraguay coinciding with fallout of
snow and rain over Patagonia. Later during the day, the wind turned SE dispersing ash over Uruguay, Brazil and
Paraguay. During 8 and 9 June, the trough intensified, shifting the ash dispersion NE throughout the trough-ridge
structure. During the first hours of 9 June, the ash cloud reached the city of Buenos Aires following a frontal
zone passing through Patagonia, leaving a thin ash layer across the area. Ash cloud continued to change in
direction over the next 6 days, with clouds following the ridge structure to the NE and SE, respectively.
Afterwards, the cloud travelled S–SE, affecting the southern part of the Patagonia and Chile.
### 4.1.1.    Regional simulation
*Model set-up*
The model domain for the regional run is presented in Table 7 and consists of 268x268 grid points covering the
northern regions of Chile and Argentina using a rotated latitude–longitude grid with a horizontal resolution of
0.15º x 0.15º and 60 vertical layers. The top pressure of the model was set to 21 hPa (∼34 km) with a mesh
refinement near the top (to capture the dispersion of ash) and the ground (to capture the characteristics of the
atmospheric boundary layer). The computational domain spans in longitude from 41º W to 81º W and in latitude
from 18º S to 58º S. Runs were performed with the on-line version of NMMB/BSC-ASH from 3 June 2011 at
00:00 UTC to 21 June 2011 at 00:00 UTC. The integration time-step for the meteorological core and aerosol
transport was set to 30 seconds. The dynamic time-steps for the long and short wave radiations were computed
every 120 time-steps. Feedback effects of ash particles on meteorology and radiation were not included in this
run. The meteorological driver was initialized with wind fields from the Era-Interim reanalysis at 0.75º x 0.75º
resolution as initial and 6-h boundary conditions. In order to reduce the errors in meteorological conditions, they
were reinitialized every 24 h with a spin-up of 12 h. Daily eruption source parameters (ESP) were obtained from
Osores et al. (2014), who estimated column heights for each eruptive pulse using the Imager Sensor data from
the GOES-13 satellite. Mass flow rate released along the column was derived from column heights based on
Mastin et al. (2009), assuming a Suzuki vertical distribution of mass typical of explosive Plinian eruptions ($A$=4 ;
$\lambda$=5). Grain size distribution was obtained from Collini et al. (2013) and discretized in 10 bins ranging from -1$\Phi$
(2 mm) to 8$\Phi$ (4 μm) with a linear dependency of particle density on diameter ranging from 1.000 to 2.200 kg m$^{-}$
$^{3}$. Particle sphericity was set to a constant standard value of 0.9 for all bins. The *percentage* aggregation model
was used to update the TGSD with a new bin for aggregates, resulting in a total of 11 bins.
*Validation of results against satellite imagery*
Model results for the airborne mass concentration of ash were validated using qualitative and quantitative
comparisons with data obtained using two different techniques. On one end, we performed a qualitative
comparison between the simulated column mass (g m$^{-2}$) from the model and the NOAA-AVHRR satellite





imagery provided by the high-resolution picture transmission (HRPT) division of the Argentinian National
Meteorological Service. Figure 3 shows how the NMMB/BSC-ASH predictions for cloud trajectory and arrival
times are in agreement with observations, capturing the three major dispersion episodes. It should be noted that
these types of images are not directly comparable because the MODIS ash detection threshold and the
reflectivity coefficients of volcanic ash are not well constrained. However, the figure illustrates the capability of
the model to predict the variation of the cloud position with time.
Column mass simulations were also validated against ash mass loadings presented by (Osores et al., 2015), who
retrieved ash-contaminated pixels detected on the basis of the concept of reverse absorption (Prata, 1989a,
1989b), i.e. those pixels with brightness temperature differences between 11 and 12 μm (BTD11-12 μm) that are
lower than 0.0 K. To minimize the presence of false positives, pixels with a BTD11-12 μm > -0.6 K and clear
sky pixels were removed. Mass loadings were mapped up to 15 g m$^{-2}$ based on an approach which combines the
satellite data with look-up tables of brightness temperatures obtained with a radiative transfer model and optical
properties of andesite volcanic rocks (Prata, 2011). Figure 4 shows a good quantitative agreement between the
model results and the airborne ash mass loadings described above.
*Validation of results against fallout deposit*
Tephra was mostly deposited eastward from the source during the first 72 h of the event within an elongated area
between 40-42º S and 64-72º W. Results from the NMMB/BSC-ASH forecast for ash deposition were validated
against: i) a detailed characterization of the proximal deposit for the first 72 h of the eruption, and ii) an isopach
map derived from measurements taken for the period beginning on 4 June until 30 June (Collini et al., 2013).
To evaluate the simulated computed thicknesses (cm) by the model near the vent during the first 72 h of the
event, model results were compared against a comprehensive classification of the proximal deposit presented by
Pistolesi et al. (2015b), who constrained the stratigraphic sequence of the deposit in different Units (phases).
Here, we constrain the deposit to the first three units of their work, corresponding to the first 72 h of the eruptive
even and including: i) Unit I, containing coarser-grained layers A-B, representing the very first stage of the
eruption within the first 50 km from the vent, and layers A–F associated to the first 24-30 h of the eruption
(afternoon of 4 to morning of 5 June); ii) Unit II, containing layer H, a fine pumice lapilli layer which was
emplaced starting on the night of 6 June; iii) Unit III, enclosing layer K2, the easiest to identify from several
coarser (fine-lapilli) grain-size layers, and being associated to the morning of 7 June. Figure 5 shows that
NMMB/BSC-ASH can reproduce the deposit presented by Pistolesi et al. (2015b) both in time and space. Key
sections located along the dispersal area (e.g. San Carlos de Bariloche – SCB, 90 km from the vent; Ingeniero
Jacobacci – IJ, 240 km east of the vent) were used as geographic references.
To evaluate the model performance at the end of our simulation, model results were also validated against an
isopach map derived from measurements taken from the 4 to 30 June presented by Collini et al. (2013). Deposit
load variations produced by remobilization were not considered in this analysis. Figure 6 shows good agreement
between the modeled deposit load (kg m$^{-2}$) at the end of the simulation and the measured ground deposit
isopachs (kg m$^{-2}$) at 30 June from Collini et al. (2013).





The model resulted in a cumulative mass of $\sim 4.2 \times 10^{11}$ kg. This value is in agreement with previous works,
where total mass was either modeled (Collini et al., 2013) or estimated by empirical fits (Bonadonna et al.,
2015b). Ashfall forecast with the NMMB/BSC-ASH model represented well the overall deposit load for the
2011 Caulle eruption.

### 4.1.2 Global simulation

For this simulation, the global domain was configured using a regular latitude–longitude grid with a horizontal
resolution of 0.75º × 1º and 60 vertical layers. The ash distribution is simulated between 3-21 June 2011 using
the Era-Interim reanalysis at 0.75º x 0.75º resolution as initial and 6-h boundary conditions. Meteorological
conditions for the global runs were also reinizialized every 24h. The atmospheric model's fundamental time-
step was set to 180 s, while the rest of the model variables remained the same as in the regional simulation.
Figure 7 shows the global dispersal of ash for the 2011 Cordon Caulle eruption after different times of the
simulation. As it can be inferred from this figure, by 10 June, the plume entered the Australian and New Zealand
airspace (Fig 7b) covering more than half of the southern hemisphere. At that point, the Civil Aviation Authority
of New Zealand warned pilots that the ash cloud was between 20,000 and 35,000 feet (6 to 11 kilometers), the
average cruising level for many aircraft (Sommer, 2011). Before the end of our simulation, on 13 June the ash
cloud had completed its first circle around the globe. This is in agreement to satellite images reported by the
Darwin Volcanic Ash Advisory Centre (Darwin VAAC, 2011). Finally, results from the global simulation are
also in agreement with those from our regional run.

### 4.1.3 Forecasting impacts on civil aviation

NMMB/BSC-ASH can furnish values of airborne concentration at relevant flight levels (FL), defined as the
vertical altitude (expressed in hundred of feet) at standard pressure at which the ash concentration is measured.
This information is particularly important for air traffic management and can be used to decide alternative routes
to avoid an encounter with a volcanic cloud. Airborne concentration at FL050 (5,000 feet on nominal pressure) is
relevant for the determination of flight cancellations and airports closure, while concentrations at FL300 (30,000
feet) are critical to assist flight dispatchers while planning flight paths and designing alternative routes in the
presence of a volcanic eruption. The model runs as if responding to an eruptive event, i.e. we only used the semi-
quantitative data available at that time as volcanological inputs.
Figure 8 shows the airspace contamination forecasted by NMMB/BSC-ASH during the 6-7 June at flight levels
FL050 and FL300, within a latitude band between 20º S and 55º S. Model results show the volcanic cloud
twisting in different directions during that period of time, achieving critical concentration values within a wide
area east of the Andes range. On 6 June, simulation results show the volcanic cloud at high atmospheric pressure
($\sim$ 30,000 feet or 300 hPa) moving northwards, and the one at lower atmospheric pressure ($\sim$ 5,000 ft or 50 hPa)
threatening the main international airports that service the region of Buenos Aires (Fig. 8a). In the morning of 7
June, the ash cloud present at lower atmospheric pressure ($\sim$ 5,000 ft or 50 hPa) changed its direction towards the
SW, ultimately affecting part of the Patagonia and Chile (Fig. 8b), while higher ash clouds started their course
around the globe (Fig. 8c). These results suggest that the cancellation of multiple flights in several Argentinean
airports during this time was justified. It is important to point out that, for this work, our objective is not to



perform a detailed study of the Caulle eruption but to use it as a blind test to confront short-term model
predictions and semi-quantitative syn-eruptive observations.

### 4.2 The 2001 Mt. Etna eruption

Mt. Etna is the most active volcano in Europe and constitutes a continuous hazard for eastern Sicily. Since 1980,
Mt. Etna has injected large volumes of pyroclasts into the atmosphere (between $10^4$ and $10^7$ m$^3$ per event) over
more than 160 eruptive episodes (Scollo et al., 2012). The explosive activity of Mt. Etna reached its climax in
2001 and 2002–03 when two major flank eruptions occurred; both characterized by long-lasting explosive
activity (Branca and Del Carlo, 2005). The 2001 event represents a good case to evaluate the deposition
mechanisms of NMMB/BSC-ASH against the well-characterized tephra deposit reported in Scollo et al. (2007).
The explosive activity at the 2570 m vent had three main phases characterized by phreatomagmatic, magmatic
and vulcanian explosions. The eruption started with a series of phreatomagmatic explosions during the first days
of the eruption. These explosions were followed by a second eruptive phase characterized by strombolian and
Hawaiian style explosions during 19-24 July. The explosive activity continued until 6 August with a series of
vulcanian explosions. Tephra fallout associated to the explosive activity during 21-24 July represented a major
source of hazard for eastern Sicily. Flight operations were cancelled at the Catania and Reggio Calabria airports
during the 22 and 23 July (Scollo et al., 2007). A detailed chronology of the eruption can be found in Scollo et
al. (2007). Volcanic plumes were captured by the Multiangle Imaging Spectro Radiometer (MISR) on board
NASA's Terra spacecraft, and analyzed with stereo matching techniques to evaluate the height of the volcanic
aerosol with a precision of a few hundred meters (Scollo et al., 2012).
Here, we validate NMMB/BSC-ASH against the tephra deposit produced from the 2570 m vent for that period of
time, and compare the model performance against simulations results from the FALL3D model (Costa et al.,
2006; Folch et al., 2009) for the same event. FALL3D is an Eulerian model for transport and deposition of
volcanic ash particles solving a set of advection-diffusion-sedimentation equations (one equation for each
particle class) on a structured terrain following grid using a second-order Finite Differences explicit scheme. The
model is used at the Buenos Aires and Darwin Volcanic Ash Advisory Centers (VAAC) in operational forecasts.

#### 4.2.1 Regional simulation

*Model set-up*

Two regional domains were used to simulate the first phase of the 2001 eruption of Mt. Etna (Table 8). The first
domain (Regional 1), used to reconstruct the tephra deposit, consists of 101x101 grid points covering the SE
flank using a rotated latitude–longitude grid with a horizontal resolution of 0.05º x 0.05º and 60 vertical layers.
Similarly to the Cordón Caulle simulations, the top pressure of the model was set to 21 hPa (∼34 km) with a
mesh refinement near the top and ground. The computational domain spans in longitude from 12.5º E to 17.5º E,
in latitude from 35.25º N to 40.25º N. Simulation runs were performed with the on-line version of NMMB/BSC-
ASH from 21 July 2001 at 00:00 UTC to 25 July 2001 at 00:00 UTC. The integration time-step for the
meteorological core was set to 10 seconds. The meteorological driver was initialized with Era-Interim reanalysis
meteorological data at 0.75º x 0.75º resolution as initial and 6-h boundary conditions. A spin-up of 12 h was used





to prepare the meteorological conditions for run. Each daily model run was reinitialized with the corresponding
reanalysis, the NMMB/BSC-ASH tracers' output from the previous day, and the associated eruption source
parameters. Meteorological conditions were reinitialized every 24 h. The grain size distribution and eruption
source parameters were obtained from Scollo et al. (2007), who assumed a Suzuki vertical mass distribution
located at the middle of the eruption column ($A$=2; $\lambda$=1), and employed the Mastin et al. (2009) empirical
relationship to characterize the MER and the Voronoi tessellation method to obtain the grain size distribution.
Finally, sensitivity analyses were performed against the different aggregation schemes available in the model. In
all cases, the TGSD was updated with a new bin for aggregates, resulting in a total of 8 bins.
A second regional domain (Regional 2) was used to evaluate tephra dispersal between 21 and 25 of July. In this
case, the domain consisted of 201x201 grid points covering a computational domain spanning in longitude from
41º E to 81º E, in latitude from 18º S to 58º S. This domain used a coarser horizontal resolution of 0.1º x 0.1º and
60 vertical layers. The integration time-step for the meteorological core was set to 30 seconds. The rest of model
set-up was kept the same as in the first regional domain (Regional 1).
***Validation of results against fallout deposit***
At the end of the second explosive phase, a continuous tephra layer covered Etna's flanks between Giarre and
Catania (from E to S). Ash deposition results from NMMB/BSC-ASH were validated against 47 samples
collected between 25 and 26 July from measured areas on flat open spaces, where the deposit did not show any
reworking. The computed tephra dispersal and deposition from NMMB/ABSC-ASH was able to reproduce the
bilobate shape of the real deposit with the two axes oriented toward Acireale and Acicastello towns. Figure 9
compares the simulated deposit load (kg m$^{-2}$) at the end of the run against the isopachs map derived from
measurements taken from the 21-24 July (Scollo et al., 2007). The model resulted in a cumulative mass of
~$1.18 \times 10^9$ kg. This value is in agreement with the results obtained from Scollo et al. (2007).
**4.2.2    Model intercomparison: NMMB/BSC-ASH vs. FALL3D**
To validate the model performance of NMMB/BSC-ASH for its operational implementation, we compare the
tephra deposition results of the model against those of the operational FALL3D model for the reconstruction of
the 2001 Mt. Etna eruption. For this comparison we ran both models using the same meteorological and
volcanological initial conditions (Table 8). Figure 10 shows the simulated thicknesses (vertical axis) for both
transport models against the observations (horizontal axis) presented in Scollo et al. (2007. The model improved
the tephra distribution results from FALL3D simulations for the same event ($R^2$; 0.80/0.62), reducing the RMSE
(0.014/0.24) and bias (0.02/0.6) and the computational time by an order of magnitude. In particular, all values
simulated with NMMB/BSC-ASH plot inside the region between 5 and 1/5 (dashed orange line) times the
observed mass at each station. The greatest differences observed against the observations for both models belong
to those points located at distances less than 15 km from the vent associated to the uncertainty in the ESPs. The
mean value of the relative error between the computed values and observed data is 64%, which improves those
from FALL3d (91%), and are comparable with those of Scollo et al., (2007), who obtained a 57% by deposit
best-fitting using the HAZMAP dispersion model





**5      Operational forecast with NMMB/BSC-ASH**
Employing on-line models for operational dispersal forecast requires larger computational resources and is not
always feasible at all operational institutes. Nevertheless, due to the increase in computing power of modern
systems, one can argue that such gradual migration towards stronger on-line coupling of NWPMs with TDMs
poses a challenging but attractive perspective from the scientific point of view for the sake of both high-quality
meteorological and volcanic ash forecasting.  The focus on volcanic aerosols integrated systems in operational
forecast is timely, since recent research has shown that meteorology and chemistry feedbacks are important in
the context of many research areas and applications (e.g., NWP and air quality forecasting, climate and Earth
system modeling (Baklanov et al., 2014)).
In that context, the Barcelona Supercomputing Center is currently working on a modeling integrated system to
provide operational forecast of volcanic ash with NMMB/BSC-ASH. The system includes a preprocessing tool
(prepares the model for real-data simulations), an executable file to run the model, and a user-based
postprocessing utility tool. Figure 11 shows a simple schematic representation of the operational implementation
of NMMB/BSC-ASH. The outcomes of this modeling system are currently being evaluated against two
operational models: i) the NOAA/ARL Hybrid Single Particle Lagrangian Integrated Trajectory Model
(HYSPLIT; Draxler and Hess, 1998) – used at the Washington VACC; and FALL3D (Costa et al., 2006; Folch
et al., 2009) – used at the Buenos Aires and Darwin VAACs. This section introduces the structure of the
operational NMMB/BSC-ASH system. Preliminary results for the model intercomparison against FALL3D are
described in Sect. 4.2.2.
**5.2      The preprocessing system**
The preprocessing utility system consists of a set of programs whose collective role is to prepare the model for
real-data simulations. Programs are grouped to preprocess geographical, meteorological and climatological
inputs and interpolate them to the model grid(s). The preprocessing system uses three main programs: *runfix*,
*degrib* and *runvariable*.
•   *Runfix* defines the model domain(s) and interpolates static geographical data to the model grid(s). In
addition to computing the latitude and longitude of the rotated grid points, this program interpolates soil
categories, land use types, terrain height, annual mean deep soil temperature, monthly albedo,
maximum snow albedo, and slope category.
•   *Degrib* extracts the necessary meteorological fields from GRIB-formatted files, used as initial condition
for global simulations and as initial and boundary conditions for single regional domains (i.e. not nested
with a global domain). GRIB files contain time-varying meteorological fields obtained from another
regional or global NWPM. In addition to the available NCEP's North American Model (NAM) or
Global Forecast System (GFS) model, the program has been updated to include European Centre for
Medium-Range Weather Forecasts (ECMWF) ERA-Interim reanalysis data (Dee et al., 2011) as
forcing.
•   *Runvariable* interpolates the meteorological fields extracted by *debgrib* to the model grid(s) defined by
*runfix* and prepares the climatological schemes. This program generates the initial and boundary





conditions that are ingested by NMMB using the NOAA Environmental Modeling System (NEMS;
Janjic, 2005; Janjic and Black, 2007), a high performance software superstructure and infrastructure
based on the Earth System Modeling Framework (ESMF) for use in operational prediction models at
NCEP.

## 5    5.3    BSC-ASH I/O files

The model takes three run-specific input files:
• The model input file (*nmmb.inp*), which defines the computational and physical schemes needed by the
8       meteorological core, the atmospheric model's fundamental time-step, and the parameterization for
9       chemical processes and radiative schemes for aerosol tracers (including ash), amongst other properties
of the model. For long-lasting eruptions, the model performs restart runs initializing the tracers from the
previous day's history file.
• The ash input file (*ash.inp*), which defines the source term. The user-defined parameters include the
13       eruption source type, column height and determination of the mass eruption rate, eruption duration,
14       gravity current conditions, aggregation processes, and particle settling velocity model. In the event of
15       various eruptive phases, the respective ESPs for each phase can be defined. Finally, the file also defines
16       the coupling strategy (on vs. off-line) employed by the model.

• The granulometry input file (*ash.tgsd*), which specifies the diameter, density, sphericity, and relative
18       mass fraction of each particle bin. This information is typically obtained from field data or created by
19       external utility programs for idealized grain size distributions. If aggregation is active, a new bin class
20       for aggregates is added to the granulometry input file.

Once a simulation is concluded, NMMB/BSC-ASH writes the following output files:
• A log file (*ash.log*), containing information about the run, including a summary of the computed
23       volcanic ash source and mass balance statistics for each time-step, and errors and warnings if any.

• A forecast results file (*problemname.nc*) in NetCDF format containing, amongst other variables, the
25       total column mass concentration ($g\ m^{-2}$) and ground deposition ($kg\ m^{-2}$) for all bins, the concentration at
26       different Flight Levels ($g\ m^{-3}$) and the Aerosol Optical Depth. This information can be processed using
27       several open-source programs to generate plots and animations. Alternatively, the post-process utility
28       program *NMMB2GMT* has been developed to generate basic GMT scripts automatically.

• A restart file (*nmmb.hst*) used to initiate a new run using the ash concentrations from a previous
30       simulation.

## 31    5.4    The Postprocess system

The postprocess utility tools are designed to interpolate outputs from the NMMB/BSC-ASH native grid(s) to
National Weather Service (NWS) standard levels (pressure, height, etc.) and standard output grids (Lambert
Conformal, polar-stereographic, etc.) in NetCDF format. The system also includes the *NMMB2GMT* program,
which uses the Generic Mapping Tools (GMT) software (Wessel and Smith, 1991) to produce similar plots to
the Volcanic Ash Graphics (VAG) used by Volcanic Ash Advisory Centers in operational forecasts.



**5.5 Scalability analysis**
To optimize a future operational implementation of the model, we aim to minimize the time-to-solution avoiding
communication overhead. In this context, we evaluate the model scalability (scaling efficiency) for its regional
and global configurations by performing a strong scalability test, in which the problem size of our simulation
(e.g., model domain and resolution) remains fixed while increasing the number of processing cores. Figure 12
shows the parallel speed-up ($S$; Eq. 19), and efficiency ($E$; Eq. 19) of the NMMB/BSC-ASH system for a global
simulation of the climatic phase for the 2011 Cordón Caulle (Table 7). On the MareNostrum-III supercomputer,
maximum efficiency for the global simulation described in Table 7 is reached at $\approx 32$ nodes (16 CPUs each) with
a parallel efficiency of 0.6.
The scalability analysis was performed on all the available source term and sedimentation schemes in the model.
The relative computational cost associated with the main processes in NMMB/BSC-ASH is presented in Fig.
13. Processes represented include: meteorological prediction, volcanic ash transport and sedimentation forecast,
aggregation of particles, gravity current effects, and the restart phase. The restart phase represents the CPU time
employed to rerun the preprocess system every 24h of simulation. This figure suggests that the computational
increase (CPU time) associated to the ash module can vary from 5 to 55%, depending on the number of
computational nodes employed. It is important to note that, depending on the settling velocity model employed,
up to 60% of the time allocated to the ash module is assigned to the sedimentation term.
Results from the scalability analysis show that the model performance (in terms of speed-up) depends on the
problem size as well as on the domain partitioning topology. In that context, the relative computational cost of
the model's meteorological core (NMMB) is evaluated as a function of its domain decomposition (e.g.,
distribution of processing units for the horizontal domains – nodes $i$ and $j$). For this analysis the bin-performance
dependency of the model is considered, therefore evaluating only the cost of one bin of ash. Results from this
analysis suggest that, for an optimal simulation using 32 nodes, the computational cost of the meteorological
core decreases over 10 % when the weight of the decomposition is focused on the $j$ nodes (e.g., more
computational resources assigned for the Fast Fourier Transformation algorithm). The best domain
decomposition resulted in 6($i$)x84($j$)+8($w$), being the number of writing processors.
For operational purposes, the computational time employed to provide ash dispersal forecast using NMMB/BSC-
ASH is evaluated for the global simulation with 1 bin of ash. The maximum time required by the model to
perform a 24 h forecast, running all the available processes (e.g., advection, diffusion, sedimentation, etc.) every
time-step (180 seconds) is less than 3 minutes when using the best domain decomposition presented before
(6x84+8). This time can be further optimized for operational purposes, i.e., calling the model physics less
frequently in order to save computational time. As a general rule of thumb, the adjustment time-step in seconds
for the meteorological core can be taken as 2.25 times the grid spacing in kilometers. For higher resolution
model runs made without parameterized convection, a time-step in seconds of about 1.9 to 2.0 times the grid
spacing may be more appropriate (Janjic and Gall, 2012).





## 6    Summary and Conclusions

We present NMMB/BSC-ASH, a new on-line multiscale meteorological and transport model developed at the Barcelona Supercomputing Center (BSC) to forecast the dispersal and deposition of volcanic aerosols. The objective of NMMB/BSC-ASH is to improve the current state-of-the-art of tephra dispersal models, especially in situations where meteorological conditions are fluctuating rapidly in time, two-way feedbacks are significant, or long-rage ash cloud dispersal predictions are necessary. The model predicts ash cloud trajectories, concentration of ash at relevant flight levels, and the expected deposit thickness for both regional and global domains. NMMB/BSC-ASH solves the mass balance equation for volcanic ash by means of a new ash module embedded in the BSC's operational system for short/mid-term chemical weather forecasts (NMMB/BSC-CTM). In addition to volcanic ash, the system is also capable to forecast the dispersion of other atmospheric aerosols (e.g. dust, sea salt, black carbon, organic aerosol, sulfates, etc.).  Its multiscale capability allows for nested global-regional atmospheric transport simulations, taking into account the characterization of the source term (emissions), the transport of volcanic particles (advection/diffusion), and the particle removal mechanisms (sedimentation/deposition).  The model has been shown to be robust and scalable to arbitrary domain sizes (regional to global) and numbers of processors.

The forecast skills of NMMB/BSC-ASH have been validated against several well-characterized eruptions, including the Pinatubo 1991 (Philippines), Etna 2001 (Italy), Chaitén 2008 (Chile) or Cordón Caulle 2011 (Chile) eruptions. To evaluate the on-line coupling strategy and the multiscale capability of the model, this paper summarizes the regional and global configurations of the model to forecast the dispersal of ash for the first days of the 2011 Cordón Caulle eruption (strong long-lasting eruption with rapid wind changes). In addition, to evaluate the sedimentation mechanisms of the model, this work also includes the results from the regional configuration of the model for the first phase of the 2001 Etna eruption, a good case study of weak long-lasting eruption with well-characterized tephra deposits. Simulation results demonstrate that NMMB/BSC-ASH is capable to reproduce the spatial and temporal dispersal variability of the ash cloud and tephra deposits.

**Software**

The work described in this paper is based on version 2.0.1 (released in April, 2014). The code, written in FORTRAN-90, is portable and efficient on available parallel computing platforms. The figures presented in this paper were generated using Gnuplot and NCAR Command Language (NCL).

**Acknowledgements**

The research leading to these results has received funding from the People Programme (Marie Curie Actions) of the European Union's Seventh Framework Programme (FP7/2007-2013) under the project NEMOH (REA grant agreement n° 289976). O. Jorba partially funded by grant CGL2013-46736 of the Ministry of Economy and Competitiveness of Spain. We are extremely grateful to the Argentinian National Meteorological Service for sharing data to validate this work; in particular we thank M.S. Osores for providing valuable insights into the eruption dynamics. Numerical simulations were performed at the Barcelona Supercomputing Center with the



MareNostrum Supercomputer using 512 and 256 - 8x4 GB DDR3-1600 DIMMS (2GB/core) Intel SandyBridge
processors, iDataPlex Compute Racks, a Linux Operating System and an Infiniband interconnection.
**Competing interests**
The authors declare that they have no conflict of interest.





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





2  **Table 1. Main characteristics of the NMMB/BSC-ASH meteorological solver.**

| *Meteorological Solver* | *Scheme* | *Reference* |
|---|---|---|
| **Spatial discretization** | | |
| Multi-scale domain ranging from large eddy simulations (LES) to global simulations | | Janjic (2005) |
| **Conservativeness** | | |
| Conservation of mass, momentum, energy, enstrophy and a number of other first order and quadratic quantities. Positive definiteness and monotonicity are preserved by tracer advection | | Janjic (1984) |
| **Coordinates /Grid** | | |
| Horizontal coordinate | Rotated latitude-longitude for regional domains, and latitude-longitude coordinate with polar filter for global domains | Janjic et al. (2009); Janjic and Gall, (2012) |
| Vertical coordinate | Terrain following hybrid sigma-pressure | Simmons and Burridge, (1981) |
| Horizontal grid | Arakawa B-grid staggering | Janjic, 2005; Janjic and Black, 2007) |
| Vertical grid | Lorenz staggering | Lorenz, (1960) |
| **Time integration schemes** | | |
| Horizontally propagating fast-waves | Forward-backward scheme | Ames, (1969); Gadd, (1974); Mesinger, (1977); Janjic, 1979) |
| Vertically propagating sound waves | Implicit scheme | Janjic and Gall, (2012) |
| Horizontal advection & Coriolis terms | Modified (Stable) Adams-Bashforth scheme | |
| Vertical advection | Crank-Nicholson scheme | Janjic, (1977,1984) |
| TKE generation and dissipation | Iterative | |
| **Advection terms** | | |
| Horizontal | Energy and enstrophy conserving, quadratic conservative, second order | Janjic and Gall, (2012) |
| Vertical | Quadratic conservative, second order | Janjic and Gall, (2012) |
| **Diffusion terms** | | |
| Vertical | Surface layer scheme | Janjic (1994, 1996) |
| Lateral | Smagorinsky non-linear approach | Janjic (1990) |
| **Physics Options** | | |
| Microphysics/Clouds | Ferrier (Eta) | Ferrier et al. (2002) |
| Short and Longwave Radiation | Rapid Radiative Transfer Model (RRTM) | Mlawer et al. (1997); Pérez et al. (2011) |
| Surface Layer | NMMB similarity theory scheme: Based on Monin-Obukhov similarity theory with Zilitinkevich thermal roughness length | Monin and Obukhov (1954); (Zilitinkevich, 1965); Janjic (1994, 1996) |
| Land Surface, Heat & moisture surface flux | LISS model | Vukovic et al. (2010) |
| Planetary Boundary layer / free atmosphere | Mellor-Yamada-Janjic scheme | Mellor and Yamada, (1982); Janjic (1996, 2001) |
| Convective adjustments | Betts-Miller-Janjic scheme | Betts and Miller, (1986); Janjic (1994, 2000). |



**Table 2. Options implemented in NMMB/BSC-ASH to estimate the mass eruption rate from column height. Unless**
**otherwise noted, the units for all parameters are in SI.**

| Reference | MER schemes | Eq. | Parameters |
|---|---|---|---|
| Mastin et al., (2009) | $MER = \bar{\rho}\left[\frac{0.5H_{plume}}{10^3}\right]^{\frac{1}{0.241}}$ | (1) | $\bar{\rho}$ = magma density (2500 kg m$^{-3}$)<br>$H_{plume}$ = column height |
| Degruyter and Bonadonna (2012) | $MER = \pi\frac{\rho_{a0}}{g'}\left(\frac{\alpha^2\bar{N}^3}{z_1^4 n}H_{plume}^4 + \frac{\beta^2\bar{N}^2\bar{v}}{6}H_{plume}^3\right)$<br>$g' = g\left(\frac{c_0\theta_0 - c_{a0}\theta_{a0}}{c_{a0}\theta_{a0}}\right)$<br>$\rho_{a0}$ =1.105, $\alpha$ =0.1, $\beta$ =0.5, $z_1$ =2.8;<br>$\theta_0$ =1200, $\theta_{a0}$ =268.7, $c0$ =1250, $ca0$ =998 | (2) | $\rho_{a0}$ = atmospheric density at the vent (kg m$^{-3}$)<br>$g'$ = reduced gravity<br>$\bar{N}$ = average buoyancy frequency (s$^{-1}$)<br>$\bar{v}$ = average wind velocity across column height ( m s$^{-1}$)<br>$z_1$ = Max. non-dimensional height<br>$\alpha, \beta$ = radial and crossflow entrainment coefficients<br>$g$ = gravitational constant (9.81 m s$^{-2}$)<br>$c_0$ = source specific heat (J kg$^{-1}$ K$^{-1}$)<br>$c_{a0}$ = specific heat of the atmosphere (J kg$^{-1}$ K$^{-1}$)<br>$\theta_0$ = source temperature (K)<br>$\theta_{a0}$ = atmospheric temperature (K) |
| Woodhouse et al. (2013) | $MER = 0.35\alpha^2 f(W_s)^4\frac{\rho_{a0}}{g'}N^3 H_{plume}^4$<br>$f(W_s) = \frac{1.44\dot{\gamma}}{\bar{N}}$<br>$g' = g\left(\frac{c_v n_0 + c_s(1-n_0)\theta_0 - c_a\theta_{a0}}{c_a\theta_{a0}}\right)$ | (3) | $Q$ = mass flux (kg s$^{-1}$)<br>$W_s$ = dimensionless wind strength<br>$\bar{N}$ = average buoyancy frequency (s$^{-1}$)<br>$\dot{\gamma}$ = shear rate of atmospheric wind (s$^{-1}$)<br>$c_s$ = specific heat of solids (J kg$^{-1}$ K$^{-1}$)<br>$c_a$ $c_s$ = specific heat of dry air (J kg$^{-1}$ K$^{-1}$) = specific heat of water vapor (J kg$^{-1}$ K$^{-1}$) |

**Table 3. Ash aggregation options in NMMB/BSC-ASH from analytical solutions based from field observations.**
**Default aggregate properties can be modified by the user.**

| Name | New aggregate class | Default properties | Reference |
|---|---|---|---|
| NONE | No aggregation processes | n/a | n/a |
| CORNELL | 50% of the 63–44 μm class aggregate<br>75% of the 44–31 μm class aggregate<br>100% of the < 31 μm class aggregate | Diameter = 250 μm<br>Density = 350 kg m$^{-3}$<br>Sphericity = 0.9 | Based on Cornell et al. (1983) Campanian Ignimbrite's deposit (Y5 ash layer) |
| PERCENTAGE | Takes a user-defined fixed percentage from each particle class | Diameter = 250 μm<br>Density = 350 kg m$^{-3}$ | Based on Sulpizio et al. (2012) |

**Table 4. Governing equations for NMMB/BSC-ASH wet aggregation model.**

| | Wet aggregation scheme | Eq. | Parameters |
|---|---|---|---|
| Number of particles of a class aggregated | $\Delta n_f \approx \frac{\Delta n_{tot}N_j}{\sum_k N_k}$  $(k = k_{min}, \ldots, k_{max})$ | (5) | $\Delta n_{tot}$ = number of particles that aggregate per time interval<br>$N_j$ = number of particles of diameter $j$ in an aggregate<br>$N_k$ = number of particles in an aggregate |



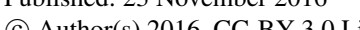

| | | | |
|---|---|---|---|
| Number of particles aggregated during $\Delta t$ | $N_j = k_f \left(\frac{d_A}{d_j}\right)^{D_f}$ | (6) | $k_f$ = fractal prefactor $\approx 1$ <br> $D_f$ = fractal exponent $\leq 3$ <br> $d_A$ = aggregate diameter <br> $d_j$ = primary particle diameter |
| Total particle decay per unit volume during $\Delta t$ | $\Delta n_{tot} = \alpha_m \left( (A_B n_{tot}^2 + A_S \emptyset^{3/D_f} n_{tot}^{2-3/D_f} + A_{DS} \emptyset^{4/D_f} n_{tot}^{2-4/D_f}) \Delta t \right)$ | (7) | $\alpha_m$ = mean sticky efficiency <br> $\emptyset$ = solid volume fraction |
| Number of particles available to aggregate | $n_{tot} = \sum_j \frac{6C_j}{\pi \rho_j d_j^3}$ | (8) | $n_{tot}$ = number of particles available to aggregate <br> $k_b$ = is the Boltzmann constant $1.38 \times 10^{-23}$ m$^2$ kg s$^{-2}$ K |
| | For Brownian motion: $A_B = -\frac{4k_b T}{3\mu_o}$ | | $T$ = absolute temperature <br> $\mu_o$ = dynamic viscosity of air |
| Kernels | Ambient fluid shear: $A_S = \frac{2\Gamma_S \varepsilon^4}{3}$ | | $\Gamma_S$ = fluid shear <br> $\varepsilon$ = particle diameter to volume fractal relationship |
| | Differential sedimentation: $A_{DS} = -\frac{\pi(\rho_m - \rho_a)g\varepsilon^4}{48\mu_o}$ | | $\rho_m$ = mean particle density <br> $\rho_a$ = air density |

2 **Table 5. Governing equations for NMMB/BSC-ASH gravity current model .**

| | *Gravity current scheme* | *Eq.* | *Parameters* |
|---|---|---|---|
| Effective radial velocity of the umbrella spreading | $u_b(t) = \frac{2}{3}\left(\frac{3\lambda Nq}{2\pi}\right)^{1/3} t^{1/3}$ <br><br> $u_b(R) = \left(\frac{2\lambda Nq}{3\pi}\right)^{1/2} \frac{1}{\sqrt{R}}$ | (9) | $u_b$ = effective radial velocity as a function of time ($t$) or cloud radius (R) <br> $\lambda$ = empirical constant ($\lambda \approx 0.2$) (Suzuki and Koyaguchi, 2009) <br> $N$ = Brunt-Väisälä frequency (atm. ambient stratification) <br> $q$ = volumetric flow rate into the umbrella region |
| Volumetric flow rate into the umbrella region | $q = C\sqrt{k}\frac{M^{3/4}}{N^{5/8}}$ <br><br> $C\begin{cases} 0.5\times10^4 \text{ m}^3\text{ kg}^{-3/4}\text{ s}^{-7/8} \\ 1.0\times10^4 \text{ m}^3\text{ kg}^{-3/4}\text{ s}^{-7/8} \end{cases}$ | (10) | $M$ = efficiency of air entrainment <br> $k$ = mass eruption rate <br> $C$ = location base constant <br> $\quad C_A$ : for tropical eruptions <br> $\quad C_B$ : for midlatitude and polar eruptions |
| Contribution of the gravitational spreading | $ct = \left(\frac{u_b}{u_b + u_w}\right) \times 100$ | (11) | $u_w$ = wind field velocity |
| Radial distance (gravity vs. passive transport) | $Ri = \frac{u_b^2}{u_w^2} = \frac{4}{9u_w^2}\left(\frac{3\lambda Nq}{2\pi}\right)^{2/3} t^{-2/3}$ | (12) | $Ri$ = Richardson number <br> $\quad Ri > 1$ : gravity-driven regime <br> $\quad Ri < 0.25$ : passive transport regime |

4 **Table 6. Settling velocity models in NMMB/BSC-ASH.**

| NAME/Reference | Drag coefficient | Eq. | Description/ Parameters |
|---|---|---|---|
| ARASTOOPOUR (Arastoopour et al., 1982) | $C_d = \begin{cases} \frac{24}{Re}\{1 + 0.15Re^{0.687}\} & Re \leq 988.947 \\ 0.44 & Re > 988.947 \end{cases}$ | (14) | For spherical particles only |



| GANSER (Ganser, 1993) | $C_d = \dfrac{24}{ReK_1}\{1 + 0.1118[Re(K_1K_2)]^{0.6567}\} + \dfrac{0.4305K_2}{1 + \dfrac{3305}{ReK_1K_2}}$ (15) | For spherical and non-spherical particles |
| | $K_1 = \dfrac{3}{[(d_n/d)] + 2\psi^{-0.5}}$ ; $K_2 = 10^{1.8148(-Log\psi)^{0.5743}}$ | $K_1, K_2$ = shape factors<br>$d_n$ = average between the min and max.<br>$d$ axis |
| | $\psi_{\text{work}} = 12.8 \dfrac{(P^2Q)^{1/3}}{1 + P(1 + Q) + 6\sqrt{1 + P^2(1 + Q^2)}}$ | $\psi_{\text{work}}$ = sphere volume<br>= particle sphericity ($\psi$ = 1 for spheres) |
| WILSON (Pfeiffer et al., 2005; Wilson and Huang, 1979) | $C_d = \begin{cases} \dfrac{24}{Re}\varphi^{-0.828} + 2\sqrt{1.07 - \varphi} & Re \leq 10^2 \\ 1 - \dfrac{1 - C_d\big|_{Re=10^2}}{900}(10^3 - Re) & 10^2 \leq Re \leq 10^3 \\ 1 & Re \leq 10^3 \end{cases}$ (16) | $\psi$ = particle aspect ratio $\psi = (b + c)2a^{-1}$<br>$a, b, c$ = particle semi-axes |
| DELLINO (Dellino et al., 2005) | $v_s = 1.2605\dfrac{v_a}{d}(Ar\varepsilon^{1.6})^{0.5206}$ (17)<br><br>$Ar = gd^3(\rho_p - \rho_a)\rho_a/\mu_a^2$ | For larger particles only<br>$Ar$ = Archimedes number<br>$g$ = gravity acceleration<br>$\varepsilon$ = particle shape factor<br>$\mu_a$ = dynamic viscosity<br>$d$ = particle equivalent diameter,<br>$\rho_p$ = particle density<br>$\rho_a$ = air density |

2 **Table 7. Model configuration for the 2011 Cordón Caulle regional and global runs. The regional run used a horizontal**
3 **resolution of 0.15º x 0.15º with a 30s dynamic time-step, while the global domain used a horizontal resolution of 1º x**
4 **0.75º with a 180s dynamic time-step.**

| MODEL CONFIGURATION | |
|---|---|
| **Dynamics** | NMMB (30s/180s time-step) |
| **Physics** | Ferrier microphysics<br>BMJ cumulus scheme<br>MYJ PBL scheme<br>LISS land surface model |
| **Aerosols** | 11 ash bins (30s/180s time-step) |
| **Source Term (emissions)** | |
| Duration | 20 days |
| Vertical distribution | Suzuki distribution |
| MER formulation | Mastin et al. (2009) |
| **Aggregation model** | Percentage |
| **Sedimentation model** | Ganser (1993) |
| **Run Set-up** | |
| Number of processors | 512 |
| Domain | Regional/Global |
| Horizontal resolution | 0.15º x 0.15º / 1º x 0.75º |
| Vertical layers | 60 |
| Top of the atmosphere | 21 hPa |
| Meteorology Boundary conditions (spatial resolutions) | ECMWF EraInterim Reanalysis  (0.75º x 0.75º) |





Table 8. Model configuration for the 2001 Mt. Etna regional simulations. Regional Run1 used a horizontal resolution of 0.1º x 0.1º with a 30s dynamic time-step, while Run2 used a finer horizontal resolution of 0.05º x 0.05º with a 10s dynamic time-step.

| | |
|---|---|
| **Source Term (emissions)** | |
| Duration | 3 days |
| Vertical distribution | Suzuki distribution |
| MER formulation | Mastin |
| Column height | 2570 |
| Ash bins | 8 |
| **Aggregation model** | Cornell et al. (1983) |
| **Sedimentation model** | Ganser (1993) |
| **Run Set-up** | |
| Number of processors | 256 |
| Domain | Regional 1 / Regional 2 |
| Horizontal resolution | 0.1º x 0.1º /  0.05º x 0.05º |
| Vertical layers | 60 |
| Top of the atmosphere | 21 hPa |
| Meteorology Boundary conditions (spatial resolutions) | ECMWF EraInterim Reanalysis  (0.75º x 0.75º) |

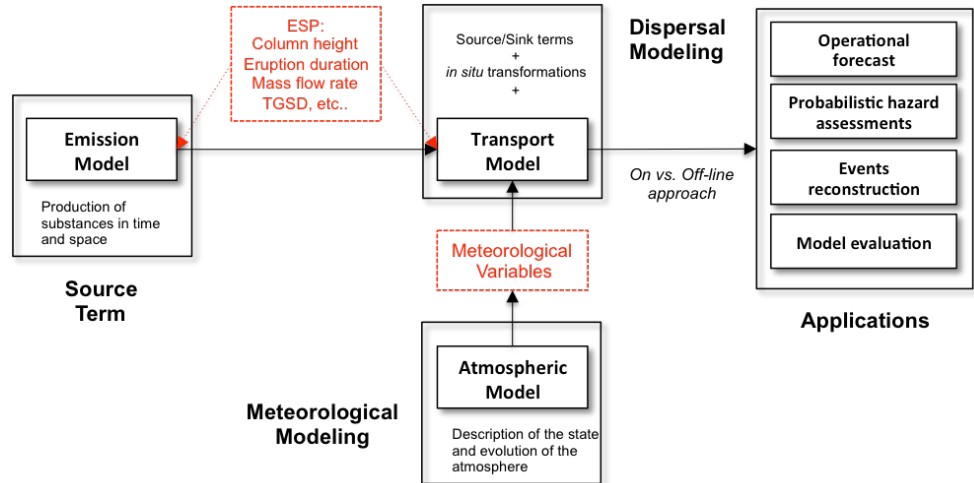

Figure 1. Schematic representation of the main components of an Atmospheric Transport Model. Red text shows model specifications for the transport of volcanic ash.





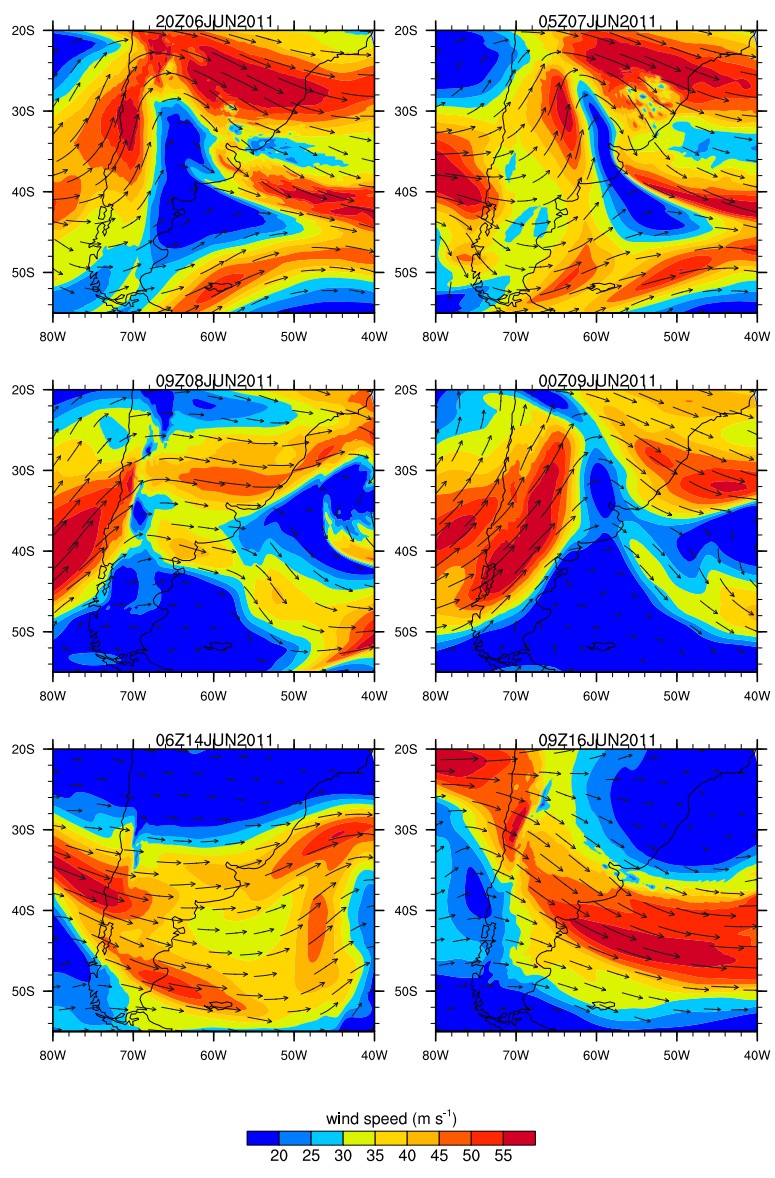

**Figure 2. Meteorological synoptic situation over Europe during 6-16 June. Plots show the direction (vector) and velocity (contours m s$^{-1}$) of the wind at 9100 m above ground level (300 hPa circa). Meteorological data obtained from the NMMB meteorological forecast driven with ERA-Interim reanalysis at 0.75° horizontal resolution.**





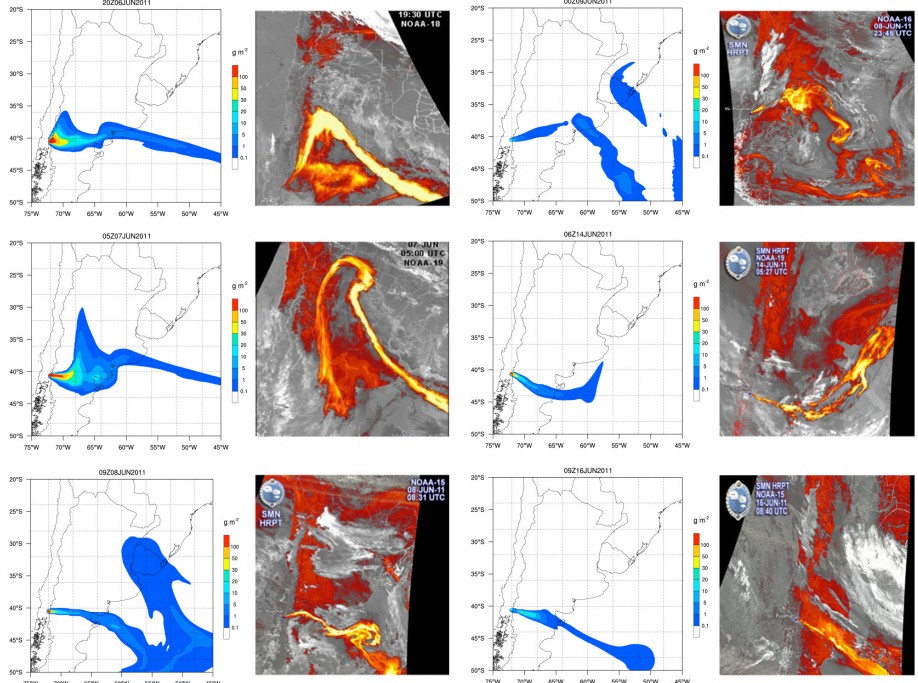

**Figure 3. Composite image of NMMB/BSC-ASH results for dispersion of ash for the 2011 Caulle eruption at different time slices. Simulation results are compared against split windows algorithm NOAA-AVHRR satellite images. Contours indicate ash column load (g m$^{-2}$) resulting from integrating the mass of the ash cloud along the atmospheric vertical levels.**





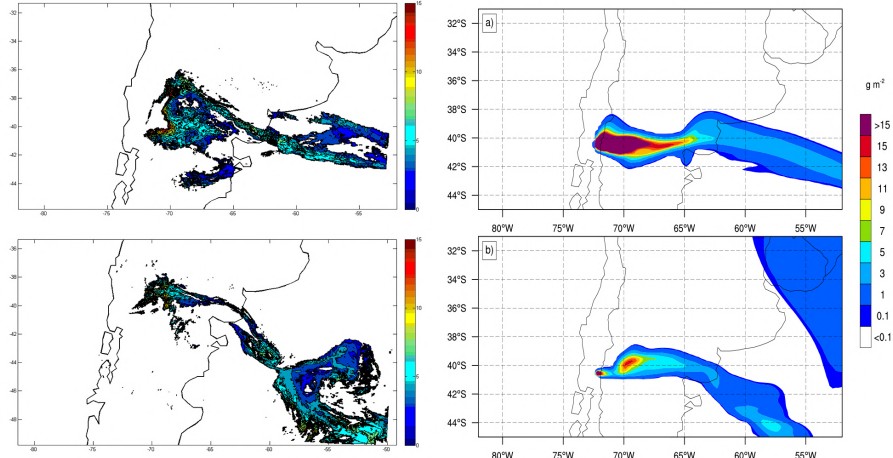

2  **Figure 4. Left: Mass loadings of the 2011 Caulle volcanic ash cloud from the MODIS-based retrievals (Osores et al.,**
3  **2015). Right: Predicted column mass (g m$^{-2}$) with NMMB/BSC-ASH for a) 6 June at 14:25 UTC and, b) 8 June at**
4  **14:15 UTC.**





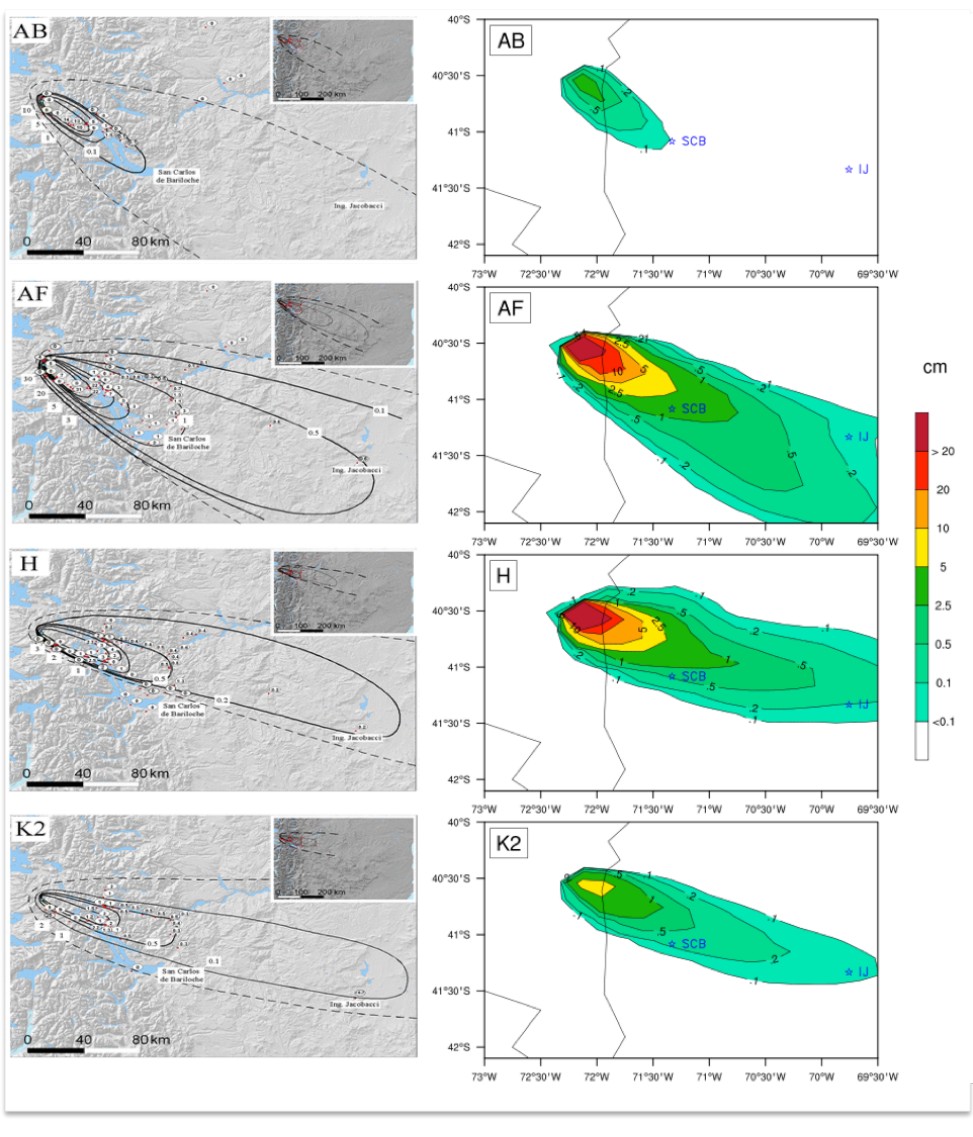

Figure 5. Left: Isopach maps in centimeter of layers A-B, A–F, H, and K2. Dashed lines infer the limit of the deposits
presented in Pistolesi et al. (2015b). Right: Corresponding NMMB/BSC-ASH computed thicknesses (cm). Key
locations in blue include San Carlos de Bariloche (SCB) and Ingeniero Jacobacci (IJ), 90 and 240 km east of the vent)



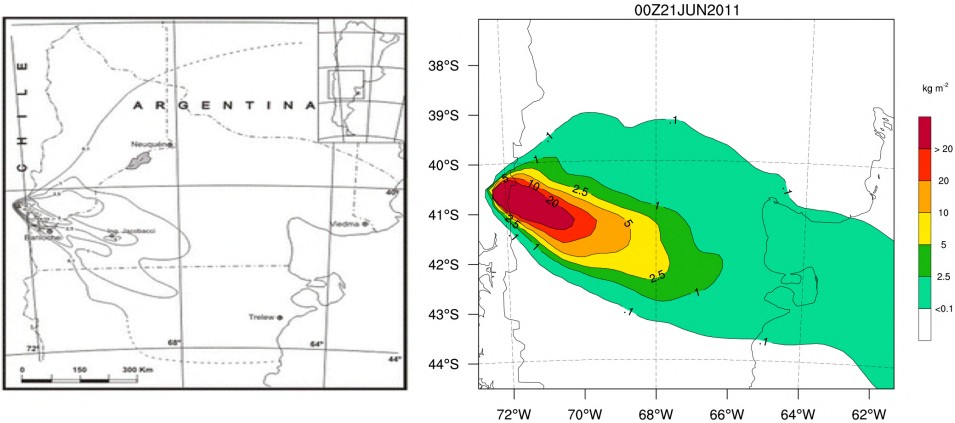

**Figure 6. Left: measured ground deposit isopachs (kg m$^{-2}$) for the period beginning on 4 June until 30 June. Dashed lines infer the limit of the deposits (Collini et al., 2013). Right: Predicted deposit load (kg m$^{-2}$) with NMMB/BSC-ASH at the end of the simulation. Key locations in blue include San Carlos de Bariloche (SCB; 90 km from the vent), Ingeniero Jacobacci (IJ; 240 km east of the vent), and Trelew and Viedma (~ 600 km SE and NE of the vent, respectively).**





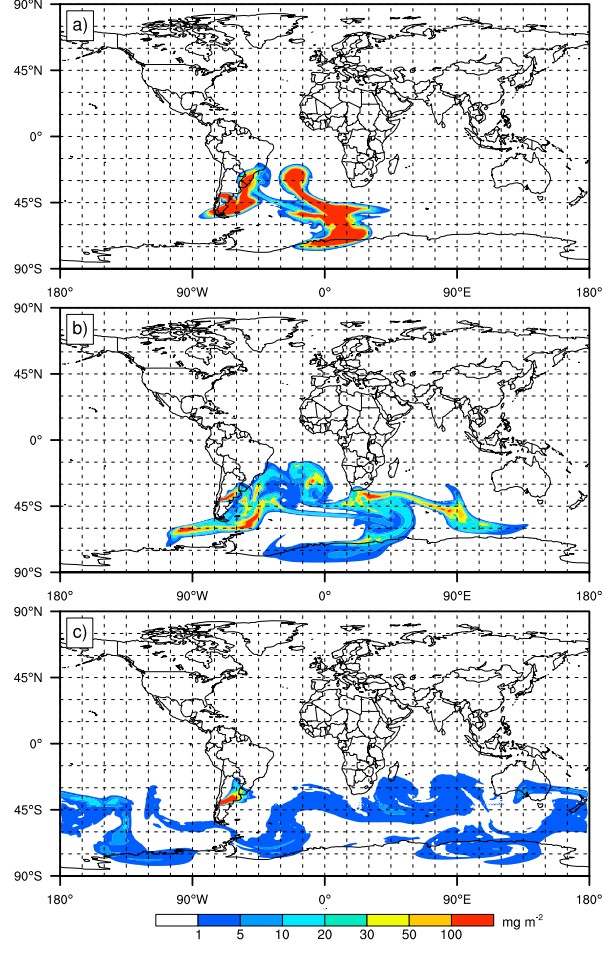

**Figure 7. NMMB/BSC-ASH total column concentration (mass loading; mg m⁻²) from our global simulation. Results for a) 8 June at 09:00 UTC, b) 10 June at 04:00 UTC, and c) 14 June at 06:00 UTC.**





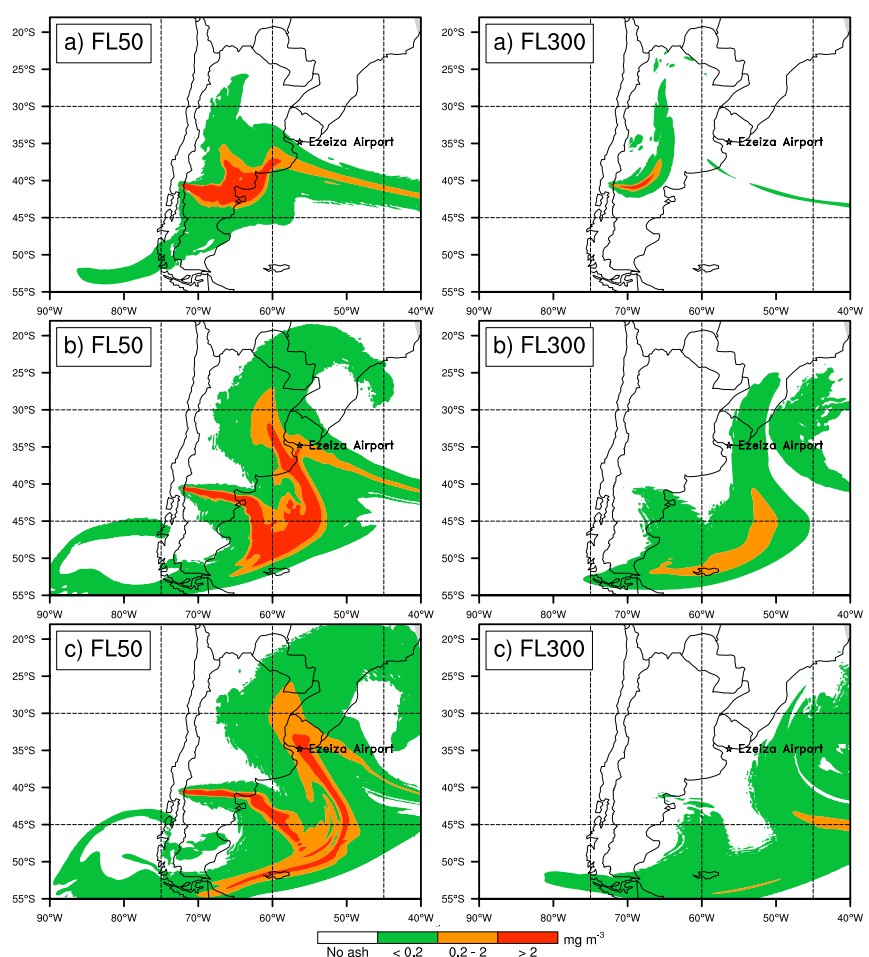

**Figure 8. NMMB/BSC-ASH Flight level ash concentrations (mass loading; mg m$^{-3}$) before and after closure of the Buenos Aires (Ezeiza) airport and air space. Results for FL50 (left) and FL300 (right) for a) 6 June at 11:00 UTC, b) 7 June at 04:00 UTC, and c) 7 June at 12:00 UTC. Safe ash concentration thresholds are shown (red contours illustrate "No Flying" zones).**





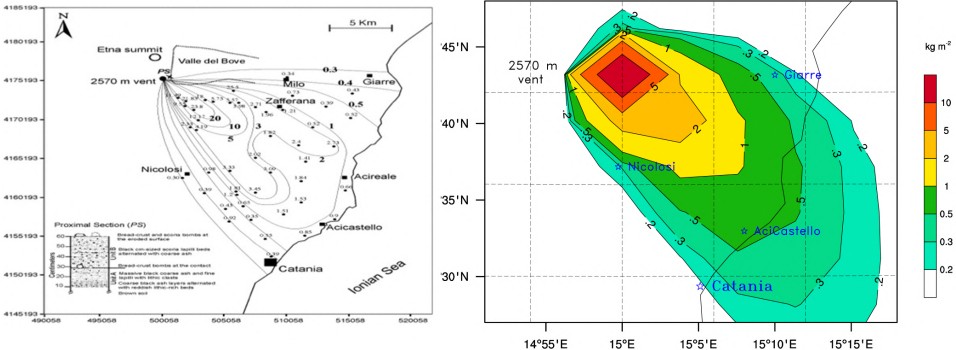

**Figure 9. Left: Isomass map of the tephra deposit formed between 21 and 24 July 2001. Curves are given in kg m⁻².**
**Coordinates are given in UTM-Datum ED50 (Scollo et al., 2007). Right: Predicted deposit load (kg m⁻²) with**
**NMMB/BSC-ASH at the end of the event.**

**Figure 10. Simulated versus observed thicknesses for the reconstruction of the 2011 Etna eruption with NMMB-ASH**
**(circles) and FALL3D (crosses). The solid bold line represents a perfect agreement, while the dashed and solid thin**
**orange lines mark the region that is different from observed thicknesses by a factor 5 (1/5) and 10 (1/10), respectively**

10   .





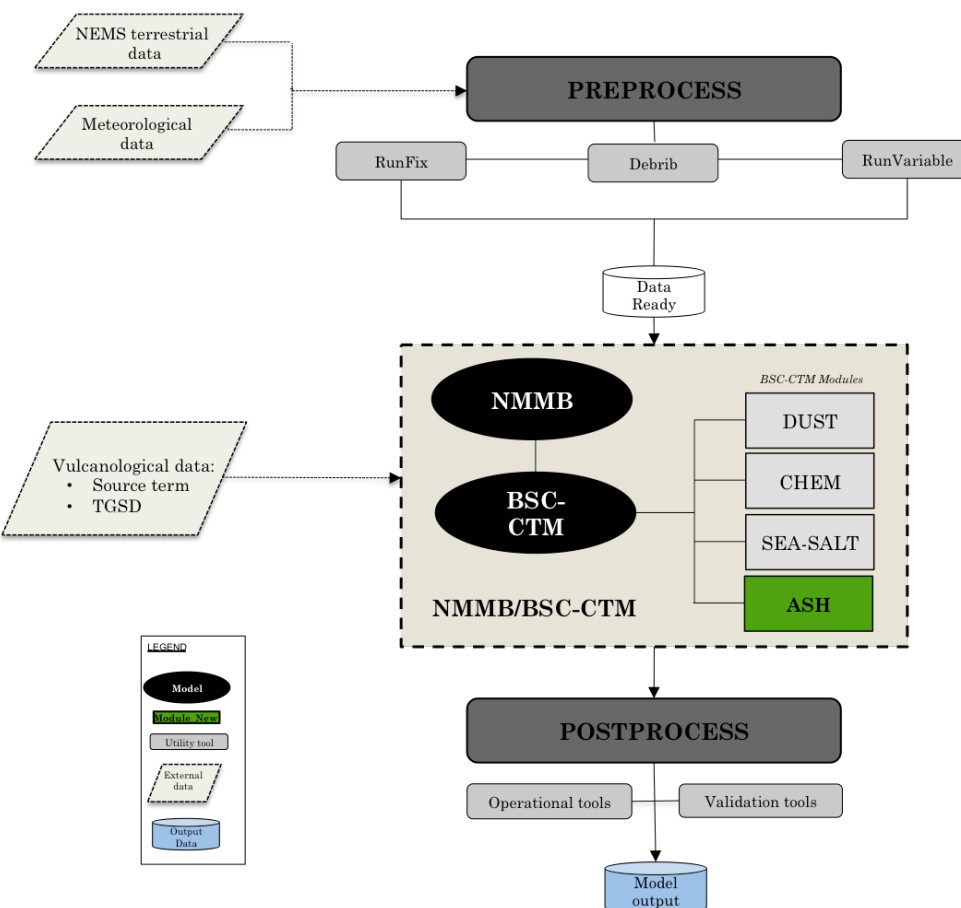

**Figure 11. Schematic representation of the operational implementation of NMMB/BSC-ASH.**



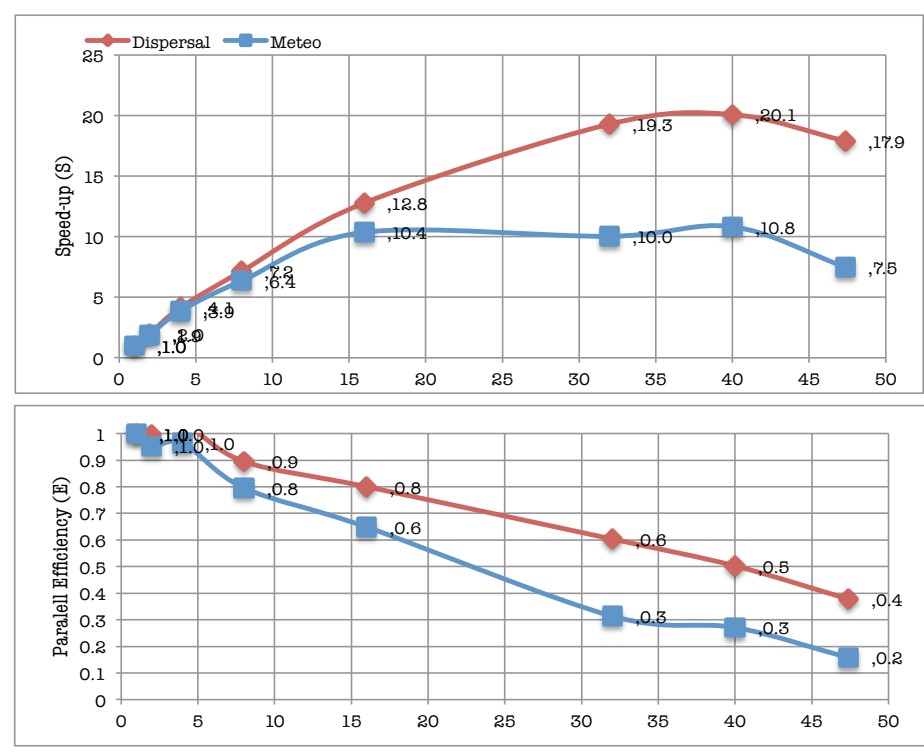

**Figure 12. Figure NMMB/BSC-ASH scalability results. Top: parallel speed-up (S; computational speed) for**
**meteorology only (blue) and for meteorology and dispersal combined (red). Bottom: parallel efficiency (E) versus**
**number of computation nodes employed.**





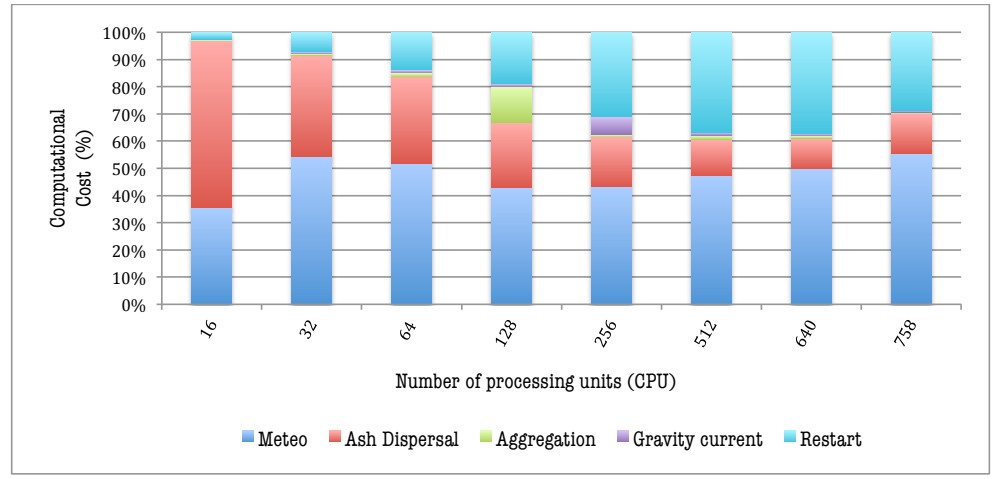

**Figure 13. NMMB/BSC-ASH relative computational cost (%) with increasing CPUs. Represented processes include:**
**Meteorology (blue); Ash dispersal for 10 bins (red); Aggregation (green); Gravity current (purple) and; Restart (light**
**blue).**