# Peer review of "Volcanic ash modeling with the on-line NMMB/BSC-ASH-v1.0 model: model description, case simulation and evaluation Alejandro Marti [(1)(\*)], Arnau Folch [(1)], Oriol Jorba [(1)] and Zavisa Janjic [(2)]"

_Atmospheric Chemistry and Physics, 2016_

## Referee Comment (RC1) · L. Mastin (Referee) · 20 Dec 2016

This paper describes the model NMMB/BSC-ASH, which solves both for meteorology and dispersion of volcanic ash. It provides to examples, from the Cordon Caulle (2011) and Etna (2001), where the model does a successful job of reproducing the meteorology and dispersion of ash clouds and deposits. And it shows that this model may be capable of running fast enough to be an operational tool, which would make it the first online operational model for volcanic ash dispersion. The paper is clearly written and presents significant advance in ash dispersion modelling. For these reasons I recommend that this paper be published. I have a couple of minor overall comments however that should be addressed.

First, It seems like the main advantage of an online model would coupling the volcanic processes with meteorology. For example, a big eruption with an umbrella cloud can modify the wind field. And eruption clouds can shade downwind areas, modifying processes like catabatic flow. But it doesn't look like you actually couple the dispersion back into the meteorology in the two examples. So, what were the advantages of this model in these cases?

Second, the physics of particle sedimentation and aggregation in Sections 3.3-3.5 and could be explained more clearly, and the equations more fully explained. Also, the equations are not numbered sequentially. Finally, some terms like monotonicity are probably understood by numerical specialists, but not by volcanologists. If you wish to attract volcanologists to the paper, I suggest you explain at least some of them. Specific ones are noted below.

These comments can be addressed with minor, technical changes to the manuscript.

Larry Mastin

Specific comments (some of which duplicate the ones above)

Page 1, Line 16: change "predicts" to "forecasts"

Page 2, line 19: Why do you call them time slabs? (rather than time slices or time intervals?). Is a slab a point in time or an interval in time?

Page 2, line 31. Is NMMB an abbreviation?

Page 3, line 7. Here you mention that the NMMB has low computational cost. Could you add a sentence or two quantifying that? Is it lower, for example, than a WRF simulation; if so, how is it more computationally more efficient? (maybe just note that you'll elaborate later in the article).

Page 3, line 38. What do you mean by a rotated latitude/longitude coordinate?

Page 4, line 17. Change "wind fields" to "wind field".
Page 4, lines 12-20. Could you explain what you mean by "effective wind fields"? Also perhaps explain the term "coupling interval" on line 19. It would also help to explain more clearly how the offline approach differs from the online approach.

Page 5, line 15. "the vertical distribution of the column shape": do you mean "the vertical distribution of mass in the column"?

Page 5, line 24. Clarify that H\_plume is the total column height above the vent (not above sea level).

Page 5, equation 4. Is S(z) in kg/s, or kg/(m s)? Seems like it should be kg/(m s), but the right-hand side of the equation appears to be in dimensions of the MER, i.e. kg/s.

Page 6, lines 27-28. "The model is based on a solution of the classical Smoluchowski equation, obtained by introducing a similarity variable and a fractal relationship for the number of primary particles in an aggregate." Is this method described in Costa et al.? If not, you might have to describe it in more detail here.

Page 7, line 13. Change "Crank-Nicholson" to "Crank-Nicolson"

Page 7, lines 17-33. It's interesting that you use the Costa et al. (2013) parameterization for radially spreading umbrella clouds. I would have thought that an online model would have the advantage of considering the momentum of umbrella spreading explicitly.

Page 8, lines 4-5. I'm a little confused by the statement that Stokes settling is considered an efficient removal mechanism for small particles (

mensionally consistent. And v\_d and v\_s are in meters per second? Are those settling velocities? Also, why are the equation numbers not sequential? They go from eq. 4 to eq. 13 to eq. 18!

Page 9, lines 6-10. Could you define monotonicity and positive definiteness? Not all readers will know what it means. Also define width halos

Page 9, line 10. Change "Nicholson" to "Nicolson".

Page 9, line 14, change "of weak long-lasting eruptions" to "of a weak, long-lasting eruption."

Page 9, beginning of Section 4. Could you say a little bit about how these eruptions could be simulated better by an online model than an offline model? Is there important coupling with the atmosphere in these cases that is not being considered with the offline model?

Page 10, line 25. Delete "over" after "spanned"

Page 10, line 26. Change "climatic" to "climactic"

Page 11, line 1. "a cloud" or "clouds"? Are you talking about the eruption cloud, or meteorological clouds?

Page 11, line 4. I'm not sure what makes this episode complementary. Perhaps just say "another episode? Did it occur at the same time as the first episode, or afterwards? At what time did it occur?

Page 11, lines 6-8. Please indicate which frame in Fig. 2 illustrates your point when describing these changes in wind.

Page 11, line 8. How is the trough illustrated in Fig. 2?

Page 11, lines 23-24: "Feedback effects of ash particles on meteorology and radiation were not included in this run". So, what is the value added using this online model?
Page 11, lines 26-28, "Daily eruption source parameters (ESP) were obtained from Osores et al. (2014), who estimated column heights for each eruptive pulse using the Imager Sensor data from the GOES-13 satellite". Could you be more specific about how height was estimated? By IR brightness temperature, assuming the cloud temperature equaled that of the surrounding atmosphere?

Page 13, lines 7-9. It seems odd that you are running the NMMB/BSC-ASH model at a horizontal resolution of 0.75x1 degree, but initializing it with ERA-interim meteorology at a horizontal resolution of 0.75x0.75 degree. What are you gaining by running the NMMB/BSC-ASH model?

Page 13, Section 4.1.2. For your global simulation, did you use all grain sizes?

Page 13, line 9. Change "reinizializated" to "reinitialized".

Page 13, line 24. Change "airports closure" to "airport closures".

Page 14, line 24. Change "terrain following grid" to "terrain-following grid". Also, change "the model is used" to "the Fall3d model is used".

Page 15, lines 32-36. It's interesting that you got better fit to the Etna data using the NMMB/BSC-ASH model than using the Fall3d model. Why do you think you got a better fit? Was the wind field produced by the NMMB/BSC-ASH model very different from that used by Fall3d? The source terms were the same for both models, right? So it had to be the wind field? Where was the wind field different? In Fig. 10, it looks to me like the fits were most improved where thicknesses were highest, and where they were lowest. Why would the NMMB/BSC-ASH model have been better in those places?

Page 17, line 14. I'm curious that you mention gravity current conditions in the source term. This is generally not considered. What do you mean by this? The existence pyroclastic flows that could serve as a source?

Page 18, line 7. Change "climatic" to "climactic".
Page 18, line 8. "maximum efficiency for the global simulation described in Table 7 is reached at  $\approx$  32 nodes". I'm having trouble seeing this in figure 12.

Page 18, line 31. What does "(6x84+8)" mean?

Page 19, line 6. Change "long-rage" to "long-range".

Page 19, lines 16-17. You say that NMMB/BSC-ASH has been validated against eruptions of Pinatubo, Etna, Chaitén, and Cordón Caulle. In this paper you only describe Etna and Cordón Caulle. Should you be citing another study for the validation against Chaitén and Pinatubo?

Table 1, equation 1. You might clarify that H\_plume is the height of the column above THE VENT (not above sea level).

Table 1, equation 2. I don't see a definition for n. Also, do you use a value of 2.8 for z1, as Degruyter and Bonadonna do?

Table 4: k and Delta-n\_f are not defined. Also, is Delta-n\_f the number of particles PER UNIT VOLUME that aggregate per unit time?

Figure 2: could you add a symbol indicating the location of Cordon Caulle volcano? Also, please label Argentine Patagonia. And add country boundaries, so we know where Paraguay is when you describe it in the text. It's also not clear why you chose these times to illustrate in this figure. It's not explained in the text or the figure caption.

Figure 3. In the satellite images, you need a scale for brightness temperature difference. And what IR bands were being differenced?

Figure 4. Mention that the color scale on the left is also g/m2. One can infer this from the text but it would be good to say it explicitly.

Figure 5. Add latitude and longitude tick marks to the left-hand maps so that they can be more directly compared with the right-hand ones.
Figure 6, left plot. Contour labels on this map are too small to read, even when enlarging the map on the computer screen.

Figure 9 caption. Perhaps change "predicted deposit load" to "modeled deposit load".

**ACPD**

---

## Referee Comment (RC2) · J. H. Sørensen (Referee) · 21 Dec 2016

General comments:

The paper presents a detailed description of the new on-line multiscale meteorological and transport model NMMB/BSC-ASH v1.0. The model, which includes two-way feedback mechanisms, is aimed at forecasting atmospheric volcanic ash dispersion. Applications of the model to two volcanic eruptions are presented.

The paper is scientifically interesting, thorough and well written. I recommend publication after minor revision.

Specific comments:

[Figure]

Interactive
comment

The fact that the NMMB/BSC-ASH model, which is intended for future operational use, involves two-way on-line coupling between meteorology and dispersion of volcanic ash is obviously an advantage and a step forward. However, the associated computational cost is probably sizable. There are large inherent uncertainties associated with forecasting dispersion of tephra, both regarding the source description and the meteorological parameters. The source model description encompasses the temporal evolution of the release of tephra, the ash column height and the initial vertical distribution of ash, all of which can fluctuate rapidly, as well as the ash particle size distribution. The uncertainty of numerical weather prediction can also be substantial with large effects on dispersion prediction, and a proper description requires use of costly ensemble prediction methods. Thus, the question is if the computational cost of carrying out two-way on-line coupling is justified against the costs of taking into account the uncertainties mentioned? I would appreciate that the authors include a related discussion of such a cost-benefit analysis.

Furthermore, I expect that the effect of on-line coupling is significant only fairly close to the eruption site, where the ash plume influences the radiation budget and the meteorological parameters. Please, comment.

The NMMB/BSC-ASH model is optimized for running on an HPC facility by employing distributed-memory parallelization (MPI). However, modern and future HPC facilities are, and will be, based on multi- or many-core processors, and thus shared-memory parallelization and thread scalability, as well as vectorization (AVX), is essential for obtaining significant performance on future HPC facilities. The authors are encouraged to comment on the model's thread scalability properties, and on possibilities for using e.g. OpenMP and OpenACL on the model code.

Technical corrections:

In addition to the many technical corrections listed by the other reviewer, Larry Mastin, I have only a single comment:

In the caption of Fig. 2, the word Europe should probably be replaced by South America.

---

## Author Comment (AC1) · 6 Feb 2017

Dear Dr. Mastin:

We appreciate your helpful comments to this paper. Your comments have been very useful to realize the need of further explanation regarding some relevant concepts presented in the manuscript. Additionally, some of your comments have prompted us to redo some of the figures. Here you will find our final answers and manuscript updates in response to your review. The revised manuscript has also been posted as a supplemental document. Your comments are in black, while our responses are in *blue italics*.

Sincerely,

Alex Marti.

**1. Dr. Mastin Review**

Page 1, Line 16: change "predicts" to "forecasts"
*Corrected- Thanks!*

Page 2, line 19: Why do you call them time slabs? (rather than time slices or time intervals?). Is a slab a point in time or an interval in time?
*Replaced time slabs for time intervals – Thanks!*

Page 2, line 31. Is NMMB an abbreviation?
*NMMB stands for Nonhydrostatic Multiscale Model on the B-grid. This abbreviation is fully spelled out in Page 3, line 1 - when describing the actual meteorological core of the model.*

Page 3, line 7. Here you mention that the NMMB has low computational cost. Could you add a sentence or two quantifying that? Is it lower, for example, than a WRF simulation; if so, how is it more computationally more efficient? (maybe just note that you'll elaborate later in the article).
*Based on the experience of NCEP, the regional version of the NMMB (meteorology only), with about equal domain and resolution, is several times (>=2) faster than the ARW.*

*Section 5.5 discusses the computational cost of NMMB/BSC-ASH and its meteorological core (e.g Fig. 13). As suggested, we have included two references to this section:*

 *- Page 3, lines 7-8: "… the low computational cost of the NMMB dynamic core presented in this work …"*
*- Page 3, line 15: "Section 5 discusses the implementation and performance of the model…".*

Page 3, line 38. What do you mean by a rotated latitude/longitude coordinate?
*We follow the Janjic (2003) methodology to rotate the longitude–latitude coordinates in the model in such a way that the coordinate origin is located in the*

*middle of the integration domain. Rotated latitude-longitude grids are employed for regional simulations in order to obtain more uniform grid distances. In this particular case, the horizontal discretization is performed on the Arakawa B-grid, with the Equator of the rotated system running through the middle of the integration domain. In this way, the reduction of the longitudinal grid-size is minimized as the southern and the northern boundaries of the integration domain are approached. Figure 1 in Janjic and Gall (2012) illustrates this regular to rotated grid transformation for a domain centered at 38N,92W.*

[Figure]

*Figure 1. Domain centered at 38N, 92W projected on: left) a regular latitude longitude map background; b) a rotated latitude longitude map background (Janjic and Gall, 2012).*

*Section 2 (Page 4, lines 1-4): we added a few sentences to describe the rotated lat-lon projection employed in the NMMB/BSC-ASH regional simulations. In addition, we added references to the two cited works above.*

Page 4, line 17. Change "wind fields" to "wind field".
*Corrected- Thanks!*

Page 4, lines 12-20. Could you explain what you mean by "effective wind fields"? Also perhaps explain the term "coupling interval" on line 19. It would also help to explain more clearly how the offline approach differs from the online approach.
*Good point. We have rewritten the introductory paragraph in Section 3 to explain more clearly the difference between the on-line and off-line model approaches, and to clarify the concept of "effective wind fields".*
*Section 3 (Page 4, lines 20-27): The on-line version of the model solves both the meteorological and aerosol transport concurrently and consistently (on-line coupling). This strategy allows the particle transport to be automatically tied to the model resolution time and space scales, resulting in a more realistic representation of the meteorological conditions. In contrast, the off-line approach uses an "effective wind field" in which, meteorological conditions (e.g. wind velocity, mid-layer pressure, etc.) are set to constant, and are only updated at specific coupling intervals (i.e. time for which meteorological fluctuations are not explicitly resolved). This strategy replicates the off-line coupling effect of traditional dispersal models used at operational levels (e.g. coupling intervals of 1h or 6h).*

Page 5, line 15. "the vertical distribution of the column shape": do you mean "the vertical distribution of mass in the column"?
*Yes; Corrected- Thanks!*

Page 5, line 24. Clarify that H_plume is the total column height above the vent (not above sea level).

*Thanks! Clarification added in Section 3.1.2 (Page 5, lines 24-25): "…: i) point source, where mass is released as a single source point at a certain height above the vent, $H_{plume}$; …"*

Page 5, equation 4. Is S(z) in kg/s, or kg/(m s)? Seems like it should be kg/(m s), but the right-hand side of the equation appears to be in dimensions of the MER, i.e. kg/s.

*Equation 4 presents the so-called Suzuki distribution of mass in the column with height, S, and is given in the dimensions of MER (i.e kg/s).*
*We have removed the term (z) in S(z) and, we have also updated the text to:*

*Section 3.1.2 (Page 5, line 28): "…where, S is the mass per unit of time (kg/s)…"*

Page 6, lines 27-28. "The model is based on a solution of the classical Smoluchowski equation, obtained by introducing a similarity variable and a fractal relationship for the number of primary particles in an aggregate." Is this method described in Costa et al.? If not, you might have to describe it in more detail here.

*The method is well described in Costa et al (2010). In their work, the authors describe a simplified model for volcanic ash aggregation, which is computationally cheaper than the full solution of the Smoluchowski [1917] equation. Section 2 in their work, describes, first the rate of change of number density of particles defined by the classical Smoluchowski equation (Eq. 1), and then their simplified version (Eq. 2) assuming the conservation of particle mass instead of particle volume and that all primary particles have the same density (i.e. information on mass size classes is transferred to the particle classes considered in the transport model).*
*We have kept the Costa et al (2010) reference for further information on the Smoluchowski equation.*

Page 7, line 13. Change "Crank-Nicholson" to "Crank-Nicolson"
*Corrected- Thanks!*

Page 7, lines 17-33. It's interesting that you use the Costa et al. (2013) parameterization for radially spreading umbrella clouds. I would have thought that an online model would have the advantage of considering the momentum of umbrella spreading explicitly.

*The observation is correct. However, the plume model embedded in NMMB/BSC-ASH (FPLUME) is a simple 1D radially-averaged model which is only one-way coupled with meteorology. To include a real 3D plume model would imply a multiphase flow simulation (CFD) near the source, which is far beyond the scope of the model.*

Page 8, lines 4-5. I'm a little confused by the statement that Stokes settling is considered an efficient removal mechanism for small particles (<20). Almost no particles of this size are removed from the atmosphere over eruptive time scales without aggregation mechanisms or rainfall scavenging.

*Correct. This paragraph was confusing and not properly expressed. The text now reads "This regime is justified for small particles and aerosols (< 20 µm) but calculating fallout times based on settling according to Stokes Law is less adequate for coarse ash (> 64 µm), which sediments much faster. In addition, ash particles are not spherical, which complicates and further slows fallout. In order to simulate properly a wider spectrum of particle sizes, NMMB/BSC-ASH adds a new sedimentation module that covers the turbulent regime"*

Page 8, line 16. Change "relaying" to "relying".
*Corrected- Thanks!*

Page 8. Equation 18 needs more explanation. What are the dimensions of R_a dn R_s? It seems like they should be in seconds per meter if this equation is to be dimensionally consistent. And v_d and v_s are in meters per second? Are those settling velocities? Also, why are the equation numbers not sequential? They go from eq. 4 to eq. 13 to eq. 18!

*Thanks for this observation. The text has been updated to define the dimensions of $R_a$ and $R_s$ $(s \cdot m^{-1})$ and $v_s$ and $v_a$ $(m \cdot s^{-1})$. In addition, we have added a sentence to clarify the terms $R_a$ and $R_s$ :*

*Section 3.3 (Page 19, lines 1-3): "These terms take into account all the effects of the lowermost layer of the atmosphere, such as turbulence ($R_a$) and Brownian diffusion, impaction and interception ($R_s$)."*

*Finally, some of the governing equations of the model are presented as Tables to enhance the flow of the text. For example, Table 6 presents equations 14 to 17. This table should be placed in Page 8 after being referenced.*

Page 9, lines 6-10. Could you define monotonicity and positive definiteness? Not all readers will know what it means. Also define width halos
*Thank you for pointing this out. The text has been updated to clarify those terms.*

*- Section 3.4 (Page 9, lines 16-17): "For these reasons, the model includes a conservative, positive definite (i.e. tracer is a positive scalar) and monotone (i.e. entirely increasing) Eulerian scheme for advection."*
*Additionally, lines 15-19 explain how these conditions are guaranteed in the model.*

*- Section 3.5 (Page 9, lines 32-33): "The Eulerian schemes in the model require relatively narrow and constant width halos (i.e. data points from the computational domain of neighboring sub-domains that are replicated locally for computational convenience), which simplify and reduce communications.*

Page 9, line 10. Change "Nicholson" to "Nicolson".
*Corrected- Thanks!*

Page 10, line 14, change "of weak long-lasting eruptions" to "of a weak, long-lasting eruption."
*Corrected- Thanks!*

Page 9, beginning of Section 4. Could you say a little bit about how these eruptions could be simulated better by an online model than an offline model? Is there important coupling with the atmosphere in these cases that is not being considered with the offline model?

*This question is in line with that of Page 4, lines 12-20 (i.e. differences on-line vs. on-line approach).*

*Text has been added at the beginning of Section 3 (Page 4, lines 20-27), to clarify why the coupling interval with the meteorology is relevant for ash dispersal forecasts. As stated in the text, off-line coupled models use an "effective wind field" in which, meteorological conditions are set to constant, and are only updated at specific coupling intervals (i.e. time for which meteorological fluctuations are not explicitly resolved). In some cases, Volcanic Ash Advisory Centers (VAACs) use off-line systems with up to 6h-coupling intervals. The inconsistencies and shortcomings of this approach (e.g. inconsistent spatial and temporal interpolation, map projections, dataset inputs, numerical schemes, etc.) could lead to errors in the ash cloud forecast (i.e. inaccurate handling of atmospheric processes with time scales smaller than the NWPM output frequency).*

*These inconsistencies may be especially important when meteorological conditions change rapidly in time, such in the case of the 2011 Caulle eruption. Therefore, the on-line approach employed in NMMB/BSC-ASH is capable to provide more accurate forecasts by removing typical inconsistencies found in traditional off-line TTDM.*

*Along with Section 3, we have updated the text in Section 4 (page 10, lines 17-18) to: ". This event represents a suitable case study of strong long-lasting eruptions with changing winds, which is useful to evaluate the advantages of the on-line approach for operational forecast."*

Page 10, line 25. Delete "over" after "spanned"
*Corrected- Thanks!*

Page 10, line 26. Change "climatic" to "climactic"
*Corrected- Thanks!*

Page 11, line 1. "a cloud" or "clouds"? Are you talking about the eruption cloud, or meteorological clouds?
*The text has been updated to clarify that we are describing the ash cloud.*

*Section 4.1 (Page 11, lines 13-14): "…The first major episode, on 4 June (18:45 UTC), resulted in an ash cloud (9-10 km) that reached…"*

Page 11, line 4. I'm not sure what makes this episode complementary. Perhaps just say "another episode? Did it occur at the same time as the first episode, or afterwards? At what time did it occur?

*Thanks for this comment. We are updating the manuscript to clarify the volcanic cloud evolution along with the synoptic meteorological situation at the time. In addition Figure 2 has been modified to expand the synoptic meteorology from 6-16 June to 4-14 June. Finally, a new reference has been added to include a recent comprehensive chronology study of the eruption* (i.e. Elissondo et al., 2016)

*Section 4.1 (page 11, lines 9-26): "Here, we describe the synoptic meteorological situation during the first two weeks of eruptive activity (Fig. 2), and give a brief chronology of the events in order to compare them with the predictions of the model. The eruption developed as a long-lasting rhyolitic activity with plume heights above the vent between around 9-10 km high a.s.l. (4-6 June), 4 and 9 km during the following week (7-14 June) and < 6 km after 14 June (Global Volcanism Program, GVP, http://www.volcano.si.edu; Siebert et al. 2010). The first major episode, on 4 June (18:45 UTC), resulted in an ash cloud (9-10 km) that reached the Chile-Argentina border within the hour of the eruption. On June 5, E-SE winds drove the plume to the Atlantic Ocean (1800 away from the source), leaving a large area of Argentina territory affected by ash fall. On June 6, the plume changed its direction abruptly toward N-NE, reaching the northern regions of the Argentine Patagonia, and then shifted direction again towards SE, threating the Buenos Aires air space. On June 7, a second episode resulted in a plume (4-9 km) dispersing ash further to the north of Argentina leading to a more recognizable shift of winds over the E-SE. On June 8, the volcanic cloud (9-10 km a.s.l.) dispersed towards NE with a bend toward SE 400 km from the source. On June 9, the plume had a NE direction reaching the city of Buenos Aires and the northern boundary of Paraguay following a frontal zone passing through Patagonia. This resulted in major air traffic disruption at the two international airports that service the city: Aeroparque (AEP) and Ezeiza (EZE), which remained closed intermittently during the following 15 days. Later during the day, the wind turned SE dispersing ash over Uruguay, Brazil and Paraguay. Ash cloud continued to change in direction over the next 6 days, with clouds following the ridge structure to the NE and SE, respectively."*

Page 11, lines 6-8. Please indicate which frame in Fig. 2 illustrates your point when describing these changes in wind.

*We have updated Fig. 2 to illustrate the daily (i.e. 1 frame per day) synoptic meteorological situation from 4 to 14 June. New text (see question above) has been included in the manuscript to describe wind changes for each day/frame (particularly from 4 to 9 June).*

Page 11, line 8. How is the trough illustrated in Fig. 2?
*See question above.*

Page 11, lines 23-24: "Feedback effects of ash particles on meteorology and radiation were not included in this run". So, what is the value added using this online model?
*That is a fair comment - thank you. Please find below an explanation of: i) the added value in NMMB/BSC-ASH from traditional tephra dispersal models; ii) the specific objective of this paper and; iii) the upcoming publications complementing this work.*

*On-line (integrated) models are defined as those where the NWPM and TDM are fully integrated in one unified modeling system using one main time-step for integration (i.e. meteorological and aerosol transport solved concurrently and consistently). The intrinsic foreseen advantages of the on-line approach employed in NMMB/BSC-ASH are:*

1. *More accurate forecasts by removing typical inconsistencies found in traditional off-line TTDM (e.g. inconsistent spatial and temporal interpolation, map projections, dataset inputs, numerical schemes, etc.).*
2. *High computational efficiency for operational forecasting of volcanic ash clouds (NMMB meteorological core is >=2x faster than WRF).*
3. *Minimal maintenance as compared to off-line models since datasets and routines are fully integrated and, therefore, must be changed in only one code.*
4. *Account for "aerosol-transport" feedbacks (one-way)*
5. *Account for "aerosols-radiative system" feedbacks (two-way).*

*These advantages are especially relevant in situations where the winds are changing rapidly with time, or the forecast is required to simulate distal ash cloud dispersal.*

*The main purpose of this paper is to provide a description of the model and to evaluate the first 4 advantages listed above. Section 4.2.2 in the manuscript illustrates how NMMB/BSC-ASH could improve traditional TTDM for operational forecasts. A more in depth study comparing On-line vs. Off-line simulations of NMMB/BSC-ASH will be available shortly (Marti et. al – in prep). This study demonstrates that meteorology-transport inconsistencies from off-line models can be, in some cases, in the same order of magnitude that those from the source term.*

*Finally, while the weather determines the transport of the emitted pollutants, their concentration, especially in the case of volcanic aerosols, influences radiative forcing and meteorological events (two-way feedbacks). Despite having a limited effect in smaller eruptions, in some cases (e.g. large explosive eruptions, super-eruptions), the impact of tropospheric volcanic aerosols can be significant, becoming a regional (or even global) radiative forcing of climate (Schmidt et al., 2015). The specific impact of volcanic aerosols on the radiative budget is currently being studied at the BSC, but is not in the scope of this paper.*

Page 11, lines 26-28, "Daily eruption source parameters (ESP) were obtained from Osores et al. (2014), who estimated column heights for each eruptive pulse using the Imager Sensor data from the GOES-13 satellite". Could you be more specific about how height was estimated? By IR brightness temperature, assuming the cloud temperature equaled that of the surrounding atmosphere?

*Column heights in Osores et al. (2014) were obtained correlating cloud-top temperatures from GOES-13 IR, assuming that the target behaves as a black body and reaches the thermal equilibrium with that of the surrounding, and with Puerto Montt thermal profile provided by daily radiosondes complementing in*

*situ observations when they were available. This technique was previously used by Swada, 1987,2002 and Holasek et al., 1996.*

*The manuscript has been updated accordantly. Section 4.1.1 (Page 12, lines 3-5): "Daily eruption source parameters (ESP) were obtained from Osores et al. (2014), who estimated column heights for each eruptive pulse using the Imager Sensor data from the GOES-13 satellite, applying the cloud-top IR image technique (Kidder and VonderHaar, 1995)".*

Page 13, lines 7-9. It seems odd that you are running the NMMB/BSC-ASH model at a horizontal resolution of 0.75x1 degree, but initializing it with ERA-interim meteorology at a horizontal resolution of 0.75x0.75 degree. What are you gaining by running the NMMB/BSC-ASH model?
*Yes, but this is true for the initial condition only. However, since NMMB/BSC-ASH is a global model, met variables are updated for transport at each model time integration step. This is not the case of driving transport with ERA-Interim, which is available only 4-times daily. So, even if the spatial resolutions are similar (or slightly coarser in our case), the temporal resolutions are not.*

Page 13, Section 4.1.2. For your global simulation, did you use all grain sizes?
*That's correct.*

*We have updated the text to clarify this. Thank you.*

*Section 4.1.2 (Page 13, lines 23-24): "…, while the rest of the model variables and grain size distribution remained the same as in the regional simulation."*

Page 13, line 9. Change "reinizializated" to "reinitialized"
*Corrected- Thanks!*

Page 13, line 24. Change "airports closure" to "airport closures"
*Corrected- Thanks!*

Page 14, line 24. Change "terrain following grid" to "terrain-following grid". Also, change "the model is used" to "the Fall3d model is used".
*Corrected- Thanks!*

Page 15, lines 32-36. It's interesting that you got better fit to the Etna data using the NMMB/BSC-ASH model than using the Fall3d model. Why do you think you got a better fit? Was the wind field produced by the NMMB/BSC-ASH model very different from that used by Fall3d? The source terms were the same for both models, right? So it had to be the wind field? Where was the wind field different? In Fig. 10, it looks to me like the fits were most improved where thicknesses were highest, and where they were lowest. Why would the NMMB/BSC-ASH model have been better in those places?

*Thank you for this comment.*
*NMMB/BSC-ASH and FALL3D used the same eruption source parameter and meteorological conditions for the 2001 Mt. Etna simulation (i.e. wind fields*

*generated with the meteorological driver in NMMB/BSC-ASH were used to drive FALL3D). There are two main reasons that explain the improved performance of NMMB/BSC-ASH:*

*1- Despite that both models use the same meteorological conditions, the on-line version of NMMB/BSC-ASH solves both the meteorological and aerosol transport concurrently and interactively at every time-step (30 sec.). FALL3D, on the other side, only allows solving the transport off-line every 1h. Changes in the wind-field conditions within that period of time leads to cumulative errors in the ash cloud forecast, especially in the distal deposit (e.g. lowest thicknesses).*

*2. While the source term module in NMMB/BSC-ASH is mainly FALL3D's, the transport of volcanic ash by advection and turbulent diffusion is analogous to those of atmospheric tracers transport (Janjic et al., 2009) in NMMB. This transport scheme uses the Adams-Bashforth scheme for horizontal advection and the Crank-Nicolson scheme for vertical advection. For the horizontal diffusion, the model uses a second order scheme with two types of parameterized dissipative processes: explicit lateral diffusion (often called horizontal diffusion, a $2^{nd}$ order nonlinear Smagorinsky-type approach; Janjic, 1990) and horizontal divergence damping (Janjic and Gall, 2012). In addition to the source term characterization, a representative particle advection during the first hours of the eruption is critical to represent the near deposit (e.g. highest thicknesses).*

*In general terms, the transport scheme employed in NMMB/BSC-ASH allows for a better representation of the transport of volcanic ash by advection and turbulent diffusion than FALL3D, resulting in better fits as demonstrated in the case of the 2001 Mt. Etna eruption.*

Page 17, line 14. I'm curious that you mention gravity current conditions in the source term. This is generally not considered. What do you mean by this? The existence pyroclastic flows that could serve as a source?

*Thank you for this comment.*
*The first part of Section 5.3 describes the different NMMB/BSC-ASH run-specific input files. In particular, we wanted to group all those parameters interacting with the ash module in a single input file (i.e. ash.inp). These parameters include, not only the characterization of the source term, but also the choice of turning on/off the gravity current model altering the particle transport in the umbrella cloud, and the coupling interval the ash input file (ash.inp).*

*To clarify the content of the ash.inp file, we have updated the text as follows – Section 5.3 (Page 17, lines 19-25): "The ash input file (ash.inp), which defines those parameters employed in the ash module. The user-defined parameters include: i) the characterization of the source term: eruption source type, column height and determination of the mass eruption rate, eruption duration, aggregation processes, and particle settling velocity model. In the event of various eruptive phases, the respective ESPs for each phase can be defined; ii) the settings to turn on/off the gravity current model altering the particle transport in the umbrella cloud; and iii) the definition of the coupling strategy (on vs. off-line) employed by the model."*

Page 18, line 7. Change "climatic" to "climactic".
*Corrected- Thanks!*

Page 18, line 8. "maximum efficiency for the global simulation described in Table 7 is reached at ≈ 32 nodes". I'm having trouble seeing this in figure 12.
*Good point. The manuscript has been updated to read: "... maximum efficiency for the global simulation described in Table 7 is reached between 32-40 nodes".*

Page 18, line 31. What does "(6x84+8)" mean?
*Another good point. Thanks. We updated the text to describe explicitly the domain decomposition.*
*Section 5.5 (Page 18, lines 34-35): "The best domain decomposition resulted in 6(i)x84(j)+8(w); where i and j, are the number of processors employed in the horizontal and vertical domains respectively, and w, the number of writing processors."*

Page 19, line 6. Change "long-rage" to "long-range"
*Corrected- Thanks!*

Page 19, lines 16-17. You say that NMMB/BSC-ASH has been validated against eruptions of Pinatubo, Etna, Chaitén, and Cordón Caulle. In this paper you only describe Etna and Cordón Caulle. Should you be citing another study for the validation against Chaitén and Pinatubo?

*Thanks for this comment. We have validated NMMB/BSC-ASH against several volcanic eruptions. The scope of this paper is to show representative eruptions for: i) a strong long-lasting eruptions with changing winds (e.g. 2011 Caulle) and ii) a weak eruption with well-characterized tephra deposits (e.g. 2001 Etna). Additional validations (e.g. Pinatubo eruption) have been presented in conferences (e.g. Marti et al., 2013, 2014). The text has been updated to include these references.*
*A complementary paper comparing the on-line and off-line strategies of the model will illustrate the model results from simulating the 2010 Eyjafjallajökull eruption.*

Table 1, equation 1. You might clarify that H_plume is the height of the column above THE VENT (not above sea level).
*Corrected- Thanks!*

Table 1, equation 2. I don't see a definition for n. Also, do you use a value of 2.8 for z1, as Degruyter and Bonadonna do?
*Thanks for this comment.*

*As indicated, $Z_1$ takes a value of 2.8* (Morton et al., 1956)*. This is actually defined, along the other constants, directly below Eq. 2.*

*However, you were right in that there was no definition for the constant n. We have defined n, as in Degruyter and Bonadonna (2012), to $\frac{1}{2^{5/2}}$ or ∼ 0.177.*

Table 4: k and Delta-n_f are not defined. Also, is Delta-n_f the number of particles PER UNIT VOLUME that aggregate per unit time?

*The table has been updated to define n and clarify Delta-n_f.*
*Delta-n_f is the Number of particles of a class aggregated per unit volume, and has been defined accordantly on the left side of the equation.*

Figure 2: could you add a symbol indicating the location of Cordon Caulle volcano? Also, please label Argentine Patagonia. And add country boundaries, so we know where Paraguay is when you describe it in the text. It's also not clear why you chose these times to illustrate in this figure. It's not explained in the text or the figure caption
*Thanks for this comment. As mentioned before we have updated Figure 2 to show the synoptic meteorological situation during the first two weeks of eruptive activity. We have also added the suggested labels and country boundaries.*
*Finally, times shown in the original Figure 2 were chosen to be consistent with Figure 3. We have modified the caption in Figure 2 to explain the choice for the new selected days (i.e. first two weeks).*

Figure 3. In the satellite images, you need a scale for brightness temperature difference. And what IR bands were being differenced?
*We have added a scale for brightness temperature as suggested. In addition, the caption of Figure 3 indicates now the IR bands being differentiated (11-12 microns)*

Figure 4. Mention that the color scale on the left is also g/m2. One can infer this from the text but it would be good to say it explicitly
*Corrected- Thanks!*

Figure 5. Add latitude and longitude tick marks to the left-hand maps so that they can be more directly compared with the right-hand ones.
*Thanks for this. We have added tick marks and labels on the left-hand maps.*

Figure 6, left plot. Contour labels on this map are too small to read, even when enlarging the map on the computer screen.
*Thanks for this. We have added tick marks and new contour labels on the left-hand plot.*

Figure 9 caption. Perhaps change "predicted deposit load" to "modeled deposit load".
*Corrected- Thanks!*

**Additional references**

- Costa, A., Folch, A. and MacEdonio, G.: A model for wet aggregation of ash particles in volcanic plumes and clouds: 1. Theoretical formulation, J. Geophys. Res. Solid Earth, 115, 1–14, doi:10.1029/2009JB007175, 2010.
- Elissondo, M., Baumann, V., Bonadonna, C., Pistolesi, M., Cioni, R., Bertagnini, A., Biass, S.,

Herrero, J. C. and Gonzalez, R.: Chronology and impact of the 2011 Cordon Caulle eruption, Chile, Nat. Hazards Earth Syst. Sci., 16(3), 675–704, doi:10.5194/nhess-16-675-2016, 2016.

- Janjic, Z.: The Step-Mountain Coordinate: Physical Package, Mon. Weather Rev., 118(7), 1429–1443, doi:10.1175/1520-0493(1990)118<1429:TSMCPP>2.0.CO;2, 1990.
- Janjic, Z.: A nonhydrostatic model based on a new approach, Meteorol. Atmos. Phys., 82, 271–285, doi:10.1007/s00703-001-0587-6, 2003.
- Janjic, Z. and Gall, R.: Scientific documentation of the NCEP nonhydrostatic multiscale model on the B grid (NMMB). Part 1 Dynamics, , (April), 72, doi:10.5065/D6WH2MZX, 2012.
- Janjic, Z., Huang, H. and Lu, S.: A unified atmospheric model suitable for studying transport of mineral aerosols from meso to global scales, IOP Conf. Ser. Earth Environ. Sci., 7, 12011, doi:10.1088/1755-1307/7/1/012011, 2009.
- Kidder, S. and VonderHaar, T.: Satellite meteorology: an introduction, Academic Press, NY., 1995.
- Marti, A., Folch, A. and Jorba, O.: On-line coupling of volcanic ash and aerosols transport with multiscale meteorological models, in IAVCEI 2013 Scientific Assembly, Kagoshima, Japan., 2013.
- Marti, A., Folch, A. and Jorba, O.: On-line coupling of volcanic ash and aerosols transport with multi-scale meteorological models, in Cities on Volcanoes 8, Jakarta, Indonesia., 2014.
- Morton, B. R., Taylor, G. and Turner, J. S.: Turbulent Gravitational Convection from Maintained and Instantaneous Sources, Proc. R. Soc. London A Math. Phys. Eng. Sci., 234(1196), 1–23 [online] Available from: http://rspa.royalsocietypublishing.org/content/234/1196/1.abstract, 1956.
- Osores, M. S., Folch, A., Ruiz, J. and Collini, E.: Estimación de alturas de columna eruptiva a partir de imáges captadas por el sensor IMAGER del GOES-13, y su empleo para el pronóstico de dispersión y depóstio de cenizas volcánicas sonre Argentina, in XIX Congreso Geologico Argentino., 2014.
- Schmidt, A., Fristad, K. and Elkins-Tanton, L.: Volcanism and Global Environmental Change, edited by A. Schmidt, K. Fristad, and L. Elkins-Tanton, Cambridge University Press., 2015.

---

## Author Comment (AC4) · 6 Feb 2017

Please find the manuscript with track changes attached

Please also note the supplement to this comment:
http://www.atmos-chem-phys-discuss.net/acp-2016-881/acp-2016-881-AC4-supplement.pdf

---

## Author Comment (AC2)

Dear Dr. Sørensen:

Thank you for your thoughtful comments to this paper. Your comments are very much in line with a complementary work to this paper, which should be ready for submission very soon. Here you will find our answers and manuscript updates in response to your review. The revised manuscript has also been posted as a supplemental document. Your comments are in black, while our responses are in *blue italics*.

Sincerely,

Alex Marti.

**2. Dr. Sørensen Review**

The fact that the NMMB/BSC-ASH model, which is intended for future operational use, involves two-way on-line coupling between meteorology and dispersion of volcanic ash is obviously an advantage and a step forward. However, the associated computational cost is probably sizable. There are large inherent uncertainties associated with forecasting dispersion of tephra, both regarding the source description and the meteorological parameters. The source model description encompasses the temporal evolution of the release of tephra, the ash column height and the initial vertical distribution of ash, all of which can fluctuate rapidly, as well as the ash particle size distribution. The uncertainty of numerical weather prediction can also be substantial with large effects on dispersion prediction, and a proper description requires use of costly ensemble prediction methods. Thus, the question is if the computational cost of carrying out two-way on-line coupling is justified against the costs of taking into account the uncertainties mentioned? I would appreciate that the authors include a related discussion of such a cost-benefit analysis.

*This is a fair question. Thanks.*

*The manuscript has been updated to accommodate a preliminary discussion regarding the cost-benefit analysis of the NMMB/BSC-ASH over traditional off-line dispersal models. A complementary study to this work is currently undergoing to quantify these benefits comparing the on-line and the off-line coupling approaches in NMMB/BSC-ASH. The magnitude of the model forecast errors implicit in the off-line approach is then compared to that of the source description.*
*Section 5.6 (Page 19, lines 10-35):*

*"Employing on-line models for operational dispersal forecast requires larger computational resources and is not always feasible at all operational institutes. Nevertheless, due to the increase in computing power of modern systems, one can argue that such gradual migration towards stronger on-line coupling of NWPMs with TDMs poses a challenging but attractive perspective from the scientific point of view for the sake of both high-quality meteorological and volcanic ash forecasting.*

*The focus on volcanic aerosols integrated systems in operational forecast is timely. Experiences from other communities (e.g. air quality) have shown the benefits from two-way online meteorology-chemistry modeling. For example, the importance of the different feedback mechanisms for meteorological and atmospheric composition processes have been previously discussed for models developed in the USA (Zhang, 2008) and Europe (Baklanov et al., 2014). These benefits have been recently stressed by several studies covering the analysis of the aerosol-transport and aerosol-radiation feedbacks onto meteorology from the air quality model evaluation international initiative (AQMEII) in its phase 2 (Alapaty et al., 2012; Galmarini et al., 2015) and the EuMetChem COST Action ES1004 (EuMetChem, http://eumetchem.info)*

*Demonstrating these benefits however, require running the on-line model with and without feedbacks over extended periods of time. For the particular case of volcanic aerosols, further research is still required to quantify the benefits posed by on-line couple models over traditional off-line TTDM on both atmospheric transport and the radiative budget. The Barcelona Supercomputing Center is currently working to quantify these benefits with the NMMB/BSC-ASH model, and assess how the magnitude of the model forecast errors implicit in the off-line approach compares with other better-constrained sources of forecast error, e.g. uncertainties in eruption source parameters. Preliminary results from this study indicate that meteorology-transport inconsistencies from off-line models can be, in some cases, in the same order of magnitude that those associated to the eruption source parameters. In terms of computational cost, the computational efficiency of the NMMB/BSC-ASH meteorological core allows for on-line integrated operational forecasts employing an equivalent computational time than FALL3D for the same computational domain and number of processing cores."*

*Finally, the feedback effects of volcanic aerosols on the radiative budget (aerosol-radiation) are currently under investigation at the BSC. However, results from other aerosol studies indicate that these feedbacks are also significant in cases where the aerosol optical depth is ≥ 3. This would be the case, for example, for strong African and Mediterranean dust intrusions (e.g Pérez et al., 2006), heat waves or fires (e.g Baró et al., 2017; Forkel et al., 2016).*

Furthermore, I expect that the effect of on-line coupling is significant only fairly close to the eruption site, where the ash plume influences the radiation budget and the meteorological parameters. Please, comment.

*Thanks for this comment. As pointed out by the reviewer, the feedback effect of volcanic aerosols on the radiative budget is especially important near the source term. However, feedback effects can also be significant for long-range transport when the aerosol optical depth is big (AOD ≥ 3). An example of this is discussed in Pérez et al. (2006), where the authors showed that, for a major dust outbreak over the Mediterranean on April 2002, the dust-radiation interaction scheme embedded into the NCEP/Eta NWP limited-area model increased accuracy for both atmospheric temperature and mean sea-level pressure forecasts across the computational domain.*

*In addition, on-line coupling systems also have significant effects in the transport of volcanic aerosols (meteorology-transport feedback). This effect is important for both proximal and long-range simulations. For the proximal deposit, a representative particle advection during the first hours of the eruption is key to represent the transport and depositions of coarse particles. For the distal deposit, on-line couple models are capable to minimize the dispersion error accumulated by off-line models (from coupling intervals; i.e. time for which meteorological fluctuations are not explicitly resolved).*

The NMMB/BSC-ASH model is optimized for running on an HPC facility by employing distributed-memory parallelization (MPI). However, modern and future HPC facilities are, and will be, based on multi- or many-core processors, and thus shared-memory parallelization and thread scalability, as well as vectorization (AVX), is essential for obtaining significant performance on future HPC facilities. The authors are encouraged to comment on the model's thread scalability properties, and on possibilities for using e.g. OpenMP and OpenACL on the model code.

*The performance analysis of a parallel code can be a challenging task. This is especially true in operational forecast where there can be multiple performance bottlenecks caused from different fields.*

*Model parallelization in NMMB/BSC-ASH is based on the well-established Message Passing Interface (MPI) library. The computational domain is decomposed into sub-domains of nearly equal size in order to balance the computational load, where each processor is in charge to solve the model equations in one sub-domain. The numerical performance and scalability of the model are presented in Section 3.5 and 5.5 in the manuscript, respectively.*

*The performance analysis of the NMMB/BSC chemical transport model has also been evaluated to identify various bottlenecks. In particular, Markomanolis et al. (2014) studied the differences between some model configurations of the model depending on the usage of extra modules. In this study they evaluated eight different topics (e.g. processor affinity, hardware encounters, domain decomposition, mapping, load imbalance issues, scalability, etc.) that could limit the scalability of the model. Their experiments used OpenMPI 1.5.4 and Intel Fortran 13.0.1. Their study identifies which computation parts of the code need to be improved and the possible reasons for the downgrade performance. Their work also illustrated, amongst other things, the importance of the processor affinity for computation intensive models and the domain decomposition across the participated nodes, and the generic load imbalance issues common for most models. The model performance could be improved through code vectorization and fix serialization procedures in the future.*

*Additional efforts are also currently undergoing to use the programming model OmpSs in order to investigate and improve the performance of the NMMB/BSC chemical transport model model. The objective here is to convert some*

*computation phases to tasks and execute them efficiently by identifying the dependencies between them. Some preliminary results have been presented here:*

*- Optimizing an Earth Science Atmospheric Application with the OmpSs Programming Model. George S. Markomanolis, Barcelona Supercomputing Center, PRACE Scientific and Industrial Conference 2014, Barcelona, Spain.*
*- Optimizing an Earth Science Atmospheric Application with the OmpSs Programming Model. G.S. Markomanolis, 16th HPC workshop on meteorology, ECMWF, Reading, UK, 2014*

In the caption of Fig. 2, the word Europe should probably be replaced by South America.
*Corrected- Thanks!*

**Additional references**

- Alapaty, K., Herwehe, J. A., Otte, T. L., Nolte, C. G., Bullock, O. R., Mallard, M. S., Kain, J. S. and Dudhia, J.: Introducing subgrid-scale cloud feedbacks to radiation for regional meteorological and climate modeling, Geophys. Res. Lett., 39(24), 1–5, doi:10.1029/2012GL054031, 2012.
- Baklanov, A., Schlünzen, K., Suppan, P., Baldasano, J. M., Brunner, D., Aksoyoglu, S., Carmichael, G., Douros, J., Flemming, J., Forkel, R., Galmarini, S., Gauss, M., Grell, G., Hirtl, M., Joffre, S., Jorba, O., Kaas, E., Kaasik, M., Kallos, G., Kong, X., Korsholm, U., Kurganskiy, A., Kushta, J., Lohmann, U., Mahura, A., Manders-Groot, A., Maurizi, A., Moussiopoulos, N., Rao, S. T., Savage, N., Seigneur, C., Sokhi, R. S., Solazzo, E., Solomos, S., Sørensen, B., Tsegas, G., Vignati, E., Vogel, B. and Zhang, Y.: Online coupled regional meteorology chemistry models in Europe: Current status and prospects, Atmos. Chem. Phys., 14(November 2013), 317–398, doi:10.5194/acp-14-317-2014, 2014.
- Baró, R., Palacios-Peña, L., Baklanov, A., Balzarini, A., Brunner, D., Forkel, R., Hirtl, M., Honzak, L., Pérez, J. L., Pirovano, G., San José, R., Schröder, W., Werhahn, J., Wolke, R., Zabkar, R. and Jiménez-Guerrero, P.: Regional effects of atmospheric aerosols on temperature: an evaluation of an ensemble of on-line coupled models, Atmos. Chem. Phys. Discuss., (January), 1–35, doi:10.5194/acp-2016-1157, 2017.
- Forkel, R., Brunner, D., Baklanov, A., Balzarini, A., Hirtl, M., Honzak, L., Jiménez-Guerrero, P., Jorba, O., Pérez, J. L., San José, R., Schröder, W., Tsegas, G., Werhahn, J., Wolke, R. and Žabkar, R.: A Multi-model Case Study on Aerosol Feedbacks in Online Coupled Chemistry-Meteorology Models Within the COST Action ES1004 EuMetChem, in Air Pollution Modeling and its Application XXIV, edited by D. G. Steyn and N. Chaumerliac, pp. 23–28, Springer International Publishing, Cham., 2016.
- Galmarini, S., Hogrefe, C., Brunner, D., Baklanov, A. and Makar, P.: Preface Article for the Atmospheric Environment Special Issue on AQMEII Phase 2, Atmos. Environ., (115), 340–344, 2015.
- Markomanolis, G. S., Jorba, O. and Baldasano, J. M.: Performance analysis of an online atmospheric-chemistry global model with Paraver: Identification of scaling limitations, Proc. 2014 Int. Conf. High Perform. Comput. Simulation, HPCS 2014, 738–745, doi:10.1109/HPCSim.2014.6903763, 2014.
- Pérez, C., Nickovic, S., Pejanovic, G., Baldasano, J. M. and Özsoy, E.: Interactive dust-radiation modeling: A step to improve weather forecasts, J. Geophys. Res. Atmos., 111(16), doi:10.1029/2005JD006717, 2006.
- Zhang, Y.: Online-coupled meteorology and chemistry models: history, current status, and outlook, Atmos. Chem. Phys., 8, 2895–2932, doi:10.5194/acp-8-2895-2008, 2008.

---

## Author Comment (AC3)

[revised manuscript text omitted]

$z_1$ = Max. non-dimensional height
$\alpha, \beta$ = radial and crossflow entrainment coefficients
$g$ = gravitational acceleration (9.81 m s$^{-2}$)
$c_0$ = source specific heat capacity (J kg$^{-1}$ K$^{-1}$)
$c_{a0}$ = specific heat capacity of the atmosphere (J kg$^{-1}$ K$^{-1}$)
$\theta_0$ = source temperature (K)
$\theta_{a0}$ = atmospheric temperature (K) |
| Woodhouse et al. (2013) | $MER = 0.35\alpha^2 f(W_s)^4\dfrac{\rho_{a0}}{g'}N^3 H_{plume}^4$

$f(W_s) = 1.44\dot{\gamma}/\bar{N}$

$g' = g\left(\dfrac{c_v n_0 + c_s(1-n_0)\theta_0 - c_a\theta_{a0}}{c_a\theta_{a0}}\right)$ | (3) | $Q$ = mass flux (kg s$^{-1}$)
$W_S$ = dimensionless wind strength
$\bar{N}$ = average buoyancy frequency (s$^{-1}$)
$\dot{\gamma}$ = shear rate of atmospheric wind (s$^{-1}$)
$c_s$ = specific heat of solids (J kg$^{-1}$ K$^{-1}$)
$c_a$ $c_s$ = specific heat of dry air (J kg$^{-1}$ K$^{-1}$)
= specific heat of water vapor (J kg$^{-1}$ K$^{-1}$) |

Table 3. Ash aggregation options in NMMB/BSC-ASH from analytical solutions based from field observations. Default aggregate properties can be modified by the user.

| Name | New aggregate class | Default properties | Reference |
|---|---|---|---|
| NONE | No aggregation processes | n/a | n/a |
| CORNELL | 50% of the 63–44 µm class aggregate
75% of the 44–31 µm class aggregate
100% of the < 31 µm class aggregate | Diameter = 250 µm
Density = 350 kg m$^{-3}$
Sphericity = 0.9 | Based on Cornell et al. (1983) Campanian Ignimbrite's deposit (Y5 ash layer) |
| PERCENTAGE | Takes a user-defined fixed percentage from each particle class | Diameter = 250 µm
Density = 350 kg m$^{-3}$ | Based on Sulpizio et al. (2012) |

Table 4. Governing equations for NMMB/BSC-ASH wet aggregation model.

| Wet aggregation scheme | | Eq. | Parameters |
|---|---|---|---|
| Number of particles of a class aggregated per unit volume | $\Delta n_f \approx \dfrac{\Delta n_{tot} N_j}{\sum_k N_k}$ $(k = k_{min}, \dots, k_{max})$ | (5) | $\Delta n_{tot}$ = number of particles that aggregate per time interval
$N_j$ = number of particles of diameter $j$ in an aggregate
$k$ = aggregation class
$N_k$ = number of particles of diameter $k$ in an aggregate |

| | | | |
|---|---|---|---|
| Number of particles aggregated during $\Delta t$ | $N_j = k_f \left(\dfrac{d_A}{d_j}\right)^{D_f}$ | (6) | $k_f$ = fractal prefactor $\approx 1$
 $D_f$ = fractal exponent $\leq 3$
 $d_A$ = aggregate diameter
 $d_j$ = primary particle diameter |
| Total particle decay per unit volume during $\Delta t$ | $\Delta n_{tot} = \alpha_m \left( (A_B n_{tot}^2 + A_S \emptyset^{3/D_f} n_{tot}^{2-3/D_f} \right.$
 $\left. + A_{DS} \emptyset^{4/D_f} n_{tot}^{2-4/D_f} \right) \Delta t$ | (7) | $\alpha_m$ = mean sticky efficiency
 $\emptyset$ = solid volume fraction |
| Number of particles available to aggregate | $n_{tot} = \sum_j \dfrac{6C_j}{\pi \rho_j d_j^3}$ | (8) | $n_{tot}$ = number of particles available to aggregate
 $k_b$ = is the Boltzmann constant $1.38 \times 10^{-23}$ m$^2$ kg s$^{-2}$ K |
| Kernels | For Brownian motion: $A_B = -\dfrac{4k_bT}{3\mu_o}$

 Ambient fluid shear: $A_S = \dfrac{2\Gamma_S \mathcal{E}^4}{3}$

 Differential sedimentation:
 $A_{DS} = -\dfrac{\pi(\rho_m - \rho_a)g\mathcal{E}^4}{48\mu_o}$ | | $T$ = absolute temperature
 $\mu_o$ = dynamic viscosity of air
 $\Gamma_S$ = fluid shear
 $\mathcal{E}$ = particle diameter to volume fractal relationship
 $\rho_m$ = mean particle density

[revised manuscript text omitted]